# Melatonin Alleviates MBP-Induced Oxidative Stress and Apoptosis in TM3 Cells via the SIRT1/PGC-1α Signaling Pathway

**DOI:** 10.3390/ijms26125910

**Published:** 2025-06-19

**Authors:** Jingjing Liu, Qingcan Guan, Shuang Li, Qi Qi, Xiaoyan Pan

**Affiliations:** Center for Reproductive Medicine, Jilin Medical University, Jilin 132013, China; liujingjing@jlmu.edu.cn (J.L.); 13196210960@163.com (Q.G.); lishuang@jlmu.edu.cn (S.L.); qiqi@jlmu.edu.cn (Q.Q.)

**Keywords:** melatonin, MBP, oxidative stress, TM3 cells, apoptosis

## Abstract

This study investigates the role of melatonin in alleviating the oxidative stress and apoptosis of TM3 Leydig cells induced by 4-methyl-2,4-bis(4-hydroxyphenyl)pent-1-ene (MBP), the primary active metabolite of Bisphenol A, and clarifies its potential mechanisms involving the SIRT1/PGC-1α pathway. We found that melatonin effectively mitigated MBP-induced cytotoxicity in TM3 cells (*p* < 0.05). The testosterone levels and steroid hormone synthesis proteins were significantly restored by melatonin. Furthermore, there was a significant reduction in apoptosis after melatonin treatment both in MBP-treated TM3 cells and Bisphenol A-treated testicular interstitial tissues (*p* < 0.05), along with a significant decrease in the pro-apoptotic markers Bax and cleaved caspase 3, and a significant increase in the anti-apoptotic Bcl-2 level and the Bcl-2/Bax ratio in TM3 cells (*p* < 0.05). Additionally, the mitochondrial membrane potential improved significantly, ROS and MDA levels were down-regulated, and ATP production was elevated following melatonin treatment in TM3 cells. Mechanistically, melatonin promoted PGC-1α expression and activated the SIRT1 signaling pathway in MBP-treated TM3 cells and Bisphenol A-treated testicular interstitial tissues. This leads to increased expression of NRF2 and its downstream antioxidant genes, mitochondrial respiratory chain complex-related genes, mitochondrial biogenesis genes, and mitochondrial fusion genes while significantly reducing mitochondrial fission genes (*p* < 0.05). The PGC-1α inhibitor SR-18292 reversed these protective effects, confirming the critical role of this pathway. Conclusively, melatonin exerts a protective effect against MBP-induced oxidative stress and apoptosis in TM3 cells through the SIRT1/PGC-1α pathway, indicating its potential as a therapeutic agent for improving male reproductive health compromised by environmental toxins.

## 1. Introduction

Male infertility has become an increasingly pressing global health issue. Approximately 15% of couples worldwide encounter difficulties in conception, with 50% of these cases attributed to male factors [1,2]. Male reproductive function is directly affected by the Leydig cells, which are essential for producing androgens and maintaining a stable environment within the seminiferous tubules [3]. Exposure to environmental toxins may damage the Leydig cells, thereby impairing male reproductive function.

Bisphenol A (BPA) is a major component of plastic products commonly encountered in daily life. Our previous study found that oral administration of BPA induced oxidative stress damage in testicular cells in mice, leading to testicular cell apoptosis and hormonal disorders [4]. Upon oral ingestion, BPA is metabolized in the liver into various metabolites, with the 4-methyl-2,4-bis(4-hydroxyphenyl)pent-1-ene (MBP) being the primary active metabolite [5]. However, it remains unclear which specific metabolites are responsible for the oxidative stress damage and apoptosis observed in testicular cells. In contrast to BPA, MBP exhibits unique structural and functional properties. It may interact with Leydig cells through mechanisms different from those of BPA. Li et al. found that 50 nM MBP significantly reduced the mRNA levels of the androgen biosynthesis genes chrome P450 11A1 (*Cyp11a1*) and *Hsd3b1* in immature Leydig cells of rats [6]. The initial and rate-limiting step of testosterone synthesis involves the transfer of cholesterol to the mitochondria by steroidogenic acute regulatory protein (StAR) [7], followed by the conversion of cholesterol to pregnenolone under the action of cytochrome P450 family members in the mitochondria, including CYP11A1 [8]. Pregnenolone is then converted to testosterone in the smooth endoplasmic reticulum. Therefore, the reduction in levels of *Cyp11a1* and *Hsd3b1* genes further inhibits androgen synthesis and negatively impacts Leydig cell function [6]. Additionally, exposure of mouse Leydig tumor cells to 1000 nM MBP resulted in the activation of nuclear factor kappa B (a transcriptional regulator of vimentin) and the demethylation of the vimentin gene promoter, which enhanced vimentin transcription and subsequently induced steroidogenesis [9]. Although MBP exposure influences the function of Leydig cells, the specific mechanisms underlying these effects remain a topic of debate.

In response to oxidative stress caused by environmental toxins, the use of antioxidants is often considered a mitigation strategy. Melatonin (MT), a hormone primarily produced by the pineal gland, possesses potent antioxidant properties that scavenge free radicals, particularly noted for its role in protecting cellular mitochondria and regulating apoptosis [10]. Recent research has demonstrated that the peroxisome proliferator-activated receptor gamma coactivator 1-alpha (PGC-1α) signaling pathway mediates the protective effects of MT against cellular oxidative stress [11,12,13]. PGC-1α is a key transcriptional co-regulator involved in mitochondrial biogenesis, metabolism, and antioxidant defense [14]. It can activate the transcription factor nuclear factor erythroid 2-related factor 2 (NRF2), induce the expression of antioxidant enzymes [15], alleviate cellular oxidative stress damage [16], and inhibit apoptosis [17]. SIRT1 (sirtuin 1) is a NAD(+)-dependent histone deacetylase that interacts with PGC-1α, leading to its deacetylation and subsequent upregulation of its transcriptional function [18,19]. MT has been shown to improve oxidative stress, mitochondrial dysfunction, and apoptosis induced by polybrominated diphenyl ethers through the SIRT1/PGC-1α signaling pathway in fish kidney cells [20] and to alleviate mitochondrial dynamics imbalance and apoptosis in BPA-induced colon injury [21]. Our previous study has also demonstrated that MT can mitigate oxidative stress damage in mouse testicular cells caused by BPA [4]. However, it remains unclear whether MT can alleviate oxidative stress damage in Leydig cells induced by MBP and the role of SIRT1/PGC-1α in this process.

This study aims to investigate the protective effects of MT against oxidative stress and apoptosis in mouse TM3 Leydig cells induced by MBP, focusing on the potential mechanisms associated with the SIRT1/PGC-1α signaling pathway. MBP was used in the in vitro experiments because it is the primary active metabolite of BPA and can directly influence cellular mechanisms, particularly in Leydig cells. Conversely, BPA was used in the in vivo experiments to reflect realistic human exposure scenarios, as BPA is commonly found in the environment and undergoes metabolic conversion to MBP within the body. By using MBP in vitro and BPA in vivo, we elucidated both the intricate cellular mechanisms influenced by the metabolite and the systemic consequences of the parent compound as encountered in real-world conditions. Our findings may deepen the understanding of MT’s biological effects on Leydig cells and its potential as a therapeutic agent for enhancing male reproductive health.

## 2. Results

### 2.1. MT Protects TM3 Cells from Toxic Effects of MBP and Alleviates Apoptosis in MBP-Treated TM3 Cells or BPA-Treated Mice via PGC-1α

#### 2.1.1. MT Reduces the Toxic Effects of MBP on TM3 Cells

The effect of MBP on the viability of TM3 cells was evaluated using the CCK-8 assay. As illustrated in Figure 1A, there were no significant differences in cell viability among 0.5 μM, 1 μM, and 5 μM MBP (*p* > 0.05). However, concentrations of 10 μM and above significantly inhibited cell viability when compared to 0.5 μM, 1 μM, and 5 μM MBP (*p* < 0.05). Notably, cell viability after treatment with 40 μM MBP was significantly lower than that of 10 μM and 20 μM MBP (*p* < 0.05). No significant difference was observed between 10 μM and 20 μM MBP (*p* > 0.05). The IC50 of MBP on TM3 cells was determined as 12.96 μM (Figure 1B), approximately 13 μM. This concentration was used in subsequent experiments.

Moreover, the CCK-8 assay also revealed that the effect of 1 μM, 10 μM, and 25 μM MT on cell viability did not yield significant results (*p* > 0.05), while 5 μM MT significantly enhanced cell viability (*p* < 0.05) (Figure 1C). Thus, the subsequent concentration of MT was established as 5 μM.

Additionally, the protective effects of MT against MBP-induced cytotoxicity were assessed through CCK-8 and LDH activity assays. After 24 h of exposure to 13 μM MBP, TM3 cells exhibited distinct morphological abnormalities, including enhanced contours, rounded shrinkage, and cellular detachment (Figure 1D). However, MT treatment resulted in considerable restoration of cell morphology and a notable increase in cell density. Statistically, cell viability in the MBP group was significantly lower than in the control group, while the MT + MBP group exhibited significantly higher cell viability compared to the MBP group (*p* < 0.05) (Figure 1E). LDH activity in the culture supernatant, indicative of cell apoptosis or death, was substantially elevated in the MBP group compared to the control group; however, the LDH activity in the MT + MBP group was significantly lower than that of the MBP group, yet remained significantly higher than that of the control group (*p* < 0.05) (Figure 1F). Therefore, MT can attenuate the toxic effects of MBP on TM3 cells.

The proliferation capacity of TM3 cells was evaluated using EdU staining, where proliferating cells were identified by red fluorescence, as shown in Figure 2A. However, no significant differences were observed in the ratio of EdU-positive cells among the control, MBP, MT + BPA, and MT + MBP + SR-18292 groups (*p* > 0.05) (Figure 2B), indicating that MT and MBP may not influence TM3 cell proliferation capacity.

To further explore the effects of MT and MBP on the secretory function of TM3 cells, ELISA was performed. The contents of StAR (Figure 3A), CYP11A1 (Figure 3B), and testosterone (Figure 3C) in TM3 cells of the MBP group were significantly lower than those observed in the control group (*p* < 0.05). In contrast, their contents in the MT + MBP group were significantly elevated compared to the MBP group (*p* < 0.05). Thus, MT can alleviate the effects of MBP on the secretory function of TM3 cells.

#### 2.1.2. MT Increases the Expression of PGC-1α in MBP-Treated TM3 Cells or Testicular Interstitial Tissue of BPA-Treated Mice

The expression of PGC-1α in TM3 cells was analyzed using Western blotting. As illustrated in Figure 4A,B, the PGC-1α expression in the MBP group was significantly lower than that in the control group (*p* < 0.05), whereas the expression in the MT + MBP group was significantly higher than in the MBP group (*p* < 0.05). There was no significant difference between the control and MT + MBP groups (*p* > 0.05). Following the administration of the PGC-1α inhibitor SR-18292, the expression levels of PGC-1α in the MT + MBP + SR-18292 group were significantly reduced compared to the MT + MBP group (*p* < 0.05), with no significant difference from the MBP group (*p* > 0.05). This indicates that MT can enhance the expression of PGC-1α in TM3 cells, which may be inhibited by the PGC-1α inhibitor.

To further verify the changes in PGC-1α expression induced by MT in vivo, we exposed mice to BPA and then detected PGC-1α expression in testicular interstitial tissue by using immunohistochemistry. As shown in Figure 4C,D, the expression of PGC-1α in testicular interstitial tissue of the BPA group was significantly lower than that in the control group, while the expression in the MT + BPA group was significantly elevated compared to the BPA group (*p* < 0.05). No significant difference was found between the control and MT + BPA groups (*p* > 0.05). These findings suggest that MT promotes the expression of PGC-1α in testicular interstitial tissue in vivo.

#### 2.1.3. MT Alleviates Apoptosis in MBP-Treated TM3 Cells or BPA-Treated Mice via PGC-1α

The TUNEL assay was conducted to assess DNA fragmentation in TM3 cells. Compared to the control group, there was a significant increase in the percentage of TUNEL-positive cells in the MBP group (*p* < 0.05) (Figure 5A,B). Conversely, the MT + MBP group exhibited a significant decrease in the percentage of TUNEL-positive cells compared to the MBP group (*p* < 0.05), with no significant difference from the control group (*p* > 0.05). The percentage of TUNEL-positive cells in the MT + MBP + SR-18292 group significantly increased compared to the MT + MBP group (*p* < 0.05) and did not significantly differ from the MBP group (*p* > 0.05).

Western blotting was performed to analyze the expression of apoptosis-related proteins Bcl-2, Bax, and cleaved caspase 3. As shown in Figure 6A–G, the expression levels of Bax and cleaved caspase 3 were significantly elevated in the MBP group, while Bcl-2 expression and the Bcl-2/Bax ratio significantly decreased, compared to the control group (*p* < 0.05). In contrast, the MT + MBP group demonstrated a significant reduction in the expression of Bax and cleaved caspase 3, accompanied by a significant increase in Bcl-2 expression and the Bcl-2/Bax ratio compared to the MBP group (*p* < 0.05). The MT + MBP + SR-18292 group exhibited a significant increase in Bax and cleaved caspase 3 levels compared to the MT + MBP group (*p* < 0.05), while Bcl-2 expression and the Bcl-2/Bax ratio significantly decreased (*p* < 0.05). Overall, SR-18292 reversed the anti-apoptotic effect of MT on TM3 cells. MT could mitigate MBP-induced apoptosis in TM3 cells via PGC-1α.

To further validate the effect of MT on apoptosis in vivo, BPA treatment was administered to mice, and TUNEL was conducted to assess cell apoptosis in testicular interstitial tissues. As illustrated in Figure 7A,B, the average fluorescence intensity of TUNEL-positive cells in the BPA group significantly increased, whereas the intensity in the MT + BPA group was substantially lower than that in the BPA group (*p* < 0.05). No significant difference was observed between the control and MT + BPA groups. This result further confirms the inhibitory effect of MT on MBP-induced apoptosis.

### 2.2. MT Alleviates MBP-Induced Oxidative Stress Damage in TM3 Cells by Activating PGC-1α/NRF2

#### 2.2.1. MT Reduces ROS and MDA Levels in TM3 Cells via PGC-1α

The levels of ROS and MDA in TM cells were assessed using DCFH-DA fluorescent probes and MDA assay kits. The green fluorescence indicated the presence of ROS (Figure 8A). Compared to the control group, the percentage of ROS-positive TM3 cells in the MBP group significantly increased (Figure 8B), accompanied by a notable rise in MDA levels (Figure 8C) (*p* < 0.05). In the MT + MBP group, the proportion of ROS-positive cells and the MDA content significantly decreased compared to the MBP group (*p* < 0.05). However, SR-18292 treatment significantly reversed the effects of MT on ROS and MDA production (*p* < 0.05). Therefore, MT may reduce the production of ROS and MDA in TM3 cells via PGC-1α.

#### 2.2.2. MT Lowers the Expression of NRF2 and Its Downstream Genes in TM3 Cells via PGC-1α

NRF2 is a crucial antioxidant stress factor regulated by PGC-1α. The expression of NRF2 protein and its downstream antioxidant genes, including NAD(P)H quinone dehydrogenase 1 (*Nqo*-*1*), superoxide dismutase 2 (*Sod2*), superoxide dismutase 1 (*Sod1*), heme oxygenase 1 (*Ho*-*1*), and glutathione peroxidase 4 (*Gpx4*), was analyzed using Western blotting and qRT-PCR in TM3 cells. The expression levels of NRF2 protein (Figure 9A,B) and the mRNA levels of *Nqo*-*1* (Figure 9C), *Sod1* (Figure 9D), *Sod2* (Figure 9E), *Ho*-*1* (Figure 9F), and *Gpx4* (Figure 9G) were significantly lower in the MBP group compared to the control group (*p* < 0.05). Notably, treatment with MT significantly increased their expression levels (*p* < 0.05). However, these effects were reversible by SR-18292. Thus, MT may enhance the expression of NRF2 and its downstream antioxidant genes via PGC-1α, exerting antioxidant effects.

### 2.3. MT Alleviates MBP-Induced Damage to Mitochondrial Homeostasis and Function in TM3 Cells via PGC-1α

#### 2.3.1. MT Mitigates MBP-Induced Damage to Mitochondrial Function in TM3 Cells via PGC-1α

Mitochondrial membrane potential (MMP) serves as an indicator of mitochondrial health. The MMP of TM3 cells was evaluated using the JC-1 probe, as shown in Figure 10A. A higher ratio of red to green fluorescence signifies an elevated MMP, suggesting improved mitochondrial health. Compared to the control group, MBP significantly decreased MMP (*p* < 0.05), while MT treatment significantly increased MMP (*p* < 0.05) (Figure 10B). However, the effect of MT was reversed by SR-18292. These results suggest that MT may enhance MMP through PGC-1α, thereby maintaining mitochondrial integrity in TM3 cells.

The levels of ATP production and the mRNA expression of the mitochondrial respiratory chain (MRC) complex-related genes in TM3 cells were assessed. The results revealed that the ATP content (Figure 11A) and mRNA levels of *Sdhb* (succinate dehydrogenase complex iron-sulfur subunit B) (Figure 11B), *Ndufb8* (NADH dehydrogenase (ubiquinone) 1 beta subcomplex subunit 8) (Figure 11C), *Cox4* (cytochrome c oxidase subunit 4) (Figure 11D), *Uqcrc2* (ubiquinol-cytochrome c reductase core protein II) (Figure 11E), and *Atp5f* (ATP synthase beta subunit precursor) (Figure 11F) in the MBP group were significantly lower than in the control group (*p* < 0.05), indicating that mitochondrial bioenergetics were compromised by MBP. Notably, MT significantly increased both ATP content and mRNA levels of MRC complex-related genes in TM3 cells (*p* < 0.05). However, this effect of MT was inhibited following treatment with SR-18292. These findings imply that MT may enhance the expression of MRC complex-related genes in TM3 cells via PGC-1α, thereby supporting mitochondrial ATP synthesis.

#### 2.3.2. MT Alleviates MBP-Induced Disruption of Mitochondrial Biogenesis and Homeostasis in TM3 Cells via PGC-1α

NRF1 and mitochondrial transcription factor A (*Tfam*) are critical factors involved in regulating mitochondrial biogenesis. The protein expression levels of NRF1 and the mRNA levels of *Tfam* in TM3 cells were analyzed. In the MBP group, the expression levels of NRF1 protein (Figure 12A,B) and *Tfam* mRNA (Figure 12C) were significantly lower than those in the control group (*p* < 0.05). Conversely, in the MT + MBP group, the expression levels of NRF1 and *Tfam* were significantly higher than those in the MBP group (*p* < 0.05), with no significant difference compared to the control group (*p* > 0.05). However, the effect of MT was inhibited by SR-18292 (*p* < 0.05), suggesting that MT may maintain mitochondrial biogenesis in TM3 cells via PGC-1α.

Mitochondrial dynamics are regulated by mitochondrial fusion and fission proteins. The mRNA expression levels of the fusion and fission-related genes were assessed using qRT-PCR, as demonstrated in Figure 13. Compared to the control group, the mRNA levels of the mitochondrial fusion genes optic atrophy 1 (*Opa1*) (Figure 13A), *Mitofusin 1* (*Mfn1*) (Figure 13B), and *Mitofusin 2* (*Mfn2*) (Figure 13C) in the MBP group significantly decreased, while the expression levels of the fission genes mitochondrial fission 1 (*Fis1*) (Figure 13D) and dynamin-related protein 1 (*Drp1*) (Figure 13E) significantly increased (*p* < 0.05). Interestingly, MT significantly enhanced the expression of fusion genes *Opa1*, *Mfn1*, and *Mfn2* while significantly reducing the expression of fission genes *Drp1* and *Fis1* (*p* < 0.05). However, these effects of MT were reversed upon treatment with SR-18292. Therefore, MT may promote mitochondrial fusion and reduce mitochondrial fission in TM3 cells via PGC-1α, thereby maintaining mitochondrial homeostasis.

### 2.4. MT Regulates the Expression of PGC-1α in TM3 Cells via SIRT1

#### 2.4.1. MT Increases SIRT1 Expression in MBP-Treated TM3 Cells or BPA-Treated Mice

TM3 cells were pretreated with MT for varying durations, and the expression of SIRT1 protein was analyzed using Western blotting. As shown in Figure 14A,B, SIRT1 protein expression significantly increased starting from the 3 h treatment compared to the 0 h and 1 h treatments (*p* < 0.05). While SIRT1 expression at 6 h and 8 h was significantly higher than at 3 h, the expression at 8 h was significantly lower than at 6 h (*p* < 0.05). The peak level of SIRT1 protein expression occurred after 6 h of MT treatment. Further analysis of the impact of MBP on SIRT1 protein expression revealed that SIRT1 expression was significantly reduced in the MBP group compared to the control group (*p* < 0.05) (Figure 14C,D). However, SIRT1 expression in the MT + MBP group significantly increased following the 6 h MT pretreatment. The SIRT1 protein inhibitor EX527 significantly decreased SIRT1 expression in the MT + MBP + EX527 group compared to the MT + MBP group (*p* < 0.05), while showing no significant difference from the MBP group (*p* > 0.05). Thus, MT mitigates the reduction in SIRT1 protein expression induced by MBP in TM3 cells.

Immunofluorescence was employed to detect SIRT1 protein expression in mouse testicular interstitial tissue. The results indicated that SIRT1 was primarily localized in Leydig cells (Figure 14E). Compared to the control group, green fluorescence intensity corresponding to SIRT1 in Leydig cells of the BPA group was significantly lower (*p* < 0.05) (Figure 14F). In contrast, the SIRT1 fluorescence intensity in the BPA + MT group was significantly higher than that in the BPA group (*p* < 0.05) and showed no significant difference from the control group (*p* > 0.05). Therefore, MT enhances SIRT1 expression in the Leydig cells of BPA-treated mice.

#### 2.4.2. Relationship Between SIRT1 and PGC-1α in TM3 Cells

To further elucidate the relationship between SIRT1 and PGC-1α, Western blotting was utilized to analyze the expression of these proteins. Compared to the control group, MBP significantly reduced the expression levels of SIRT1 (Figure 15A,B) and PGC-1α (Figure 15C,D) proteins in TM3 cells (*p* < 0.05). Following MT pretreatment, the expression of both SIRT1 and PGC-1α proteins in TM3 cells significantly increased (*p* < 0.05). However, the SIRT1 inhibitor EX-527 significantly decreased PGC-1α protein expression, while the PGC-1α inhibitor SR-18292 significantly reduced SIRT1 protein expression. These results indicate an interaction between SIRT1 and PGC-1α proteins in TM3 cells, with MT simultaneously regulating the expression of both proteins.

## 3. Discussion

The toxic effects of MBP on Leydig cells have significantly compromised male reproductive health [6,9]. The specific mechanisms through which MT mitigates MBP-induced damage to Leydig cells, particularly via the SIRT1/PGC-1α pathway, remain inadequately investigated. This study systematically elucidated the protective effects of MT against MBP-induced apoptosis and oxidative stress in TM3 cells, which mimic the Leydig cells of adult male mice. The results revealed that these effects were closely associated with the upregulation of PGC-1α expression, activation of the NRF2 antioxidant pathway, and maintenance of mitochondrial homeostasis. MT demonstrated considerable protective effects against cell damage induced by MBP, suggesting its potential as a therapeutic agent for enhancing male reproductive health.

In our in vitro culture of TM3 cells, the oxygen concentration in the cultivation environment is around 20%, which may induce mild oxidative stress compared to the in vivo environment. At concentrations from 1 μM to 5 μM, MT may exert antioxidant properties to mitigate this oxidative stress, resulting in enhanced cell viability. However, further increasing the concentration of MT to above 10 μM resulted in a reduction in cell viability. Similarly, Govindasamy et al. found that when Schwann cells were treated with MT concentrations ranging from 0.5 to 20 μM for 24 h, cell viability increased with concentrations up to 10 μM, but decreased at 20 μM [22]. This decline observed in both our study and that of Govindasamy et al. [22] could be attributed to cytotoxic effects associated with higher concentrations of MT. The differences in the optimal MT concentration for maximum cell viability may be due to the varying tolerance of different cell types to MT.

MT exhibits anti-apoptotic properties. For example, it alleviates oligodendrocyte apoptosis by regulating the expression of Bim and Bcl-2 [23], mitigates oxidative stress-induced apoptosis in ovarian granulosa cells through the promotion of mitochondrial autophagy [17], and inhibits doxorubicin-induced oxidative stress, pyroptosis, and apoptosis in cardiomyocytes by activating the SIRT1/NRF2 pathway [24]. Moreover, MT can alleviate dibutyl phthalate-induced ferroptosis in mouse Leydig cells by inhibiting the oxidative stress-induced Sp2/VDAC2 signaling pathway [25]. It can also suppress oxidative stress and apoptosis in Leydig cells via the SIRT1 signaling pathway [26]. However, the role of PGC-1α in the alleviating effects of MT on Leydig cell apoptosis remains rarely reported. PGC-1α is a transcriptional coactivator whose mechanism involves interactions with various nuclear receptors and transcription factors, enhancing their transcriptional activity and regulating processes such as mitochondrial biogenesis, energy metabolism, and thermogenesis. Abnormal expression of p53/PGC-1α can mediate mitochondrial dysfunction, promoting apoptosis in PC3 prostate cancer cells [27]. In inflammatory environments, PGC-1α can upregulate the expression of anti-apoptotic genes, alleviating TNFα or lipopolysaccharide-induced hepatocyte apoptosis [28]. Overexpression of PGC-1α in annulus fibrosus cells can prevent apoptosis and mitochondrial autophagy [29]. Our study demonstrated that MT significantly upregulated PGC-1α and Bcl-2 expression in TM3 cells while downregulating Bax and cleaved caspase 3, thereby reducing the pro-apoptotic effects of MBP. However, the PGC-1α inhibitor SR-18292 reversed the protective effect of MT against MBP-induced TM3 cell apoptosis, indicating that MT may exert its anti-apoptotic effect on TM3 cells through PGC-1α. Notably, our findings indicated that MT did not significantly affect TM3 cell proliferation capacity. Furthermore, our study found that MT enhanced the expression of steroid hormone synthesis-related proteins StAR and CYP11A1, thus promoting the synthesis and secretion of androgens. Previous studies have shown that MT can alleviate steroid hormone synthesis disorders in testicular Leydig cells caused by diabetic hyperglycemia [11], CoCl(2)-induced hypoxia [30], and BPA [3], promoting testosterone synthesis and secretion, which may be closely related to the protective effect of MT against apoptosis in Leydig cells. The enhancement of expression of StAR and CYP11A1 proteins by MT further suggests that its alleviating effect on testosterone synthesis in TM3 cells may be closely related to its protective role in mitochondrial function.

Activation of PGC-1α enhances the expression of NRF2. The NRF2 signaling pathway is widely recognized as a fundamental protective mechanism that enhances cellular antioxidant defense and mitigates oxidative stress damage [31]. Under physiological conditions, NRF2 resides in the cytoplasm and binds to Keap1 (Kelch-like ECH-associated protein 1), which inhibits NRF2 activity [32]. Under stress conditions, NRF2 is released from the NRF2-Keap1 complex and translocates to the nucleus [33], where it binds to antioxidant response elements in the promoter region and upregulates the expression of downstream antioxidant genes, such as *Nqo*-*1*, *Ho*-*1*, *Sod*, and *Gpx4* [34]. NRF2 and its downstream effectors are involved in cell growth and apoptosis [35], which may be a key mechanism in regulating oxidative stress damage to cells. In this study, MT promoted the expression of the antioxidant genes *Nqo*-*1*, *Ho*-*1*, *Sod1*, *Sod2*, and *Gpx4* by activating PGC-1α and NRF2, which may facilitate the clearance of excess ROS and maintain cellular redox balance. A decrease in the levels of ROS and MDA, along with an enhancement in ATP production, was observed in MT-treated cells, further supporting the notion that MT alleviates oxidative stress damage in TM3 cells and exerts a protective effect on mitochondrial function. The PGC-1α inhibitor SR-18292 reversed these effects, underscoring the key role of PGC-1α in this signaling cascade. The activation of the PGC-1α/NRF2 pathway by MT may be closely associated with the alleviation of oxidative stress damage in TM3 cells caused by MBP.

Mitochondria have been identified as a critical target for MT [36]. MT maintains mitochondrial homeostasis and integrity by regulating essential biological processes, including mitochondrial redox balance, bioenergetics, biogenesis, dynamics, and autophagy [37]. Recent studies suggest that MT can mitigate mitochondrial dysfunction induced by MBP [38]. Exposure to MBP induces mitochondrial stress, significantly increasing mitochondrial-derived ROS and leading to the collapse of the MMP [39]. The MMP is essential for maintaining normal mitochondrial function, established by the transfer of protons from the MRC complexes I, III, and IV to the intermembrane space, thereby creating a proton gradient across the inner membrane [40]. During oxidative phosphorylation, a normal MMP directly drives ATP production in mitochondria. A decrease in MMP, accompanied by electron leakage from the MRC, causes leaked electrons to react with O_2_, promoting ROS generation. The ROS can further damage the mitochondrial membrane, reduce the MMP, inhibit oxidative phosphorylation, and obstruct ATP production [41]. In our study, the results indicated that MBP promoted the accumulation of ROS in TM3 cells, decreased the MMP, and reduced ATP production, while MT significantly reversed these adverse effects, maintaining normal mitochondrial function.

Mitochondrial biogenesis and dynamics are key to regulating mitochondrial quality and maintaining normal ATP production. MT-induced activation of PGC-1α plays a critical role in the regulation of mitochondrial biogenesis [42]. Yu et al. demonstrated that MT reduced mitochondrial oxidative stress and promoted mitochondrial biogenesis through the AMP-activated protein kinase/PGC-1α/SIRT3 signaling pathway, thereby ensuring mitochondrial functionality [43]. A decrease in PGC-1α expression can lead to increased levels of ROS and diminished expression of transcription factors, including NRF1, resulting in various metabolic disorders and reproductive diseases [44,45,46]. NRF1 and PGC-1α can collaboratively activate the Tfam, directly promoting the replication and transcription of the mitochondrial genome as well as mitochondrial biogenesis [47]. Mitochondrial DNA encodes subunits of MRC complexes I, III, IV, and V, which are involved in mitochondrial oxidative phosphorylation and regulate ATP production [48]. Furthermore, some studies indicate that NRF2 promotes mitochondrial biogenesis by regulating PGC-1α in the nervous system [49,50]. Additionally, Wang et al. also found that MT enhanced mitochondrial biogenesis in Leydig cells induced by hyperglycemia in diabetes by activating the PGC-1α/NRF1 pathway, thereby improving the steroidogenic function of stromal cells [11]. Consistently, we found that MT significantly increased the expression of mitochondrial biogenesis-related proteins PGC-1α, NRF1, NRF2, and Tfam, as well as the expression of MRC complex genes and ATP production in TM3 cells. Thus, MT could alleviate the impact of MBP on mitochondrial biogenesis in TM3 cells.

Mitochondrial dynamics affect the exchange of mitochondrial content between individual mitochondria and the communication between mitochondria and the cytosol, thereby influencing mitochondrial physiology and cellular function [51]. Mitochondrial fusion involves the fusion of the outer and inner membranes, primarily mediated by Mfn1 and Mfn2, with regulation by Opa1. In contrast, mitochondrial fission is mediated by the receptors Drp1 and Fis1, which are located on the outer membrane [52]. Increasing evidence suggests that disturbances in mitochondrial fusion and fission lead to mitochondrial dysfunction [53,54,55]. Our study found that MBP significantly elevated the expression of mitochondrial fission genes *Drp1* and *Fis1*, while decreasing the expression of mitochondrial fusion genes *Opa1*, *Mfn1*, and *Mfn2*, disrupting mitochondrial dynamics. Excessive mitochondrial fission inhibits the activity of the MRC complexes, leading to increased ROS and mitochondrial dysfunction [42,56]. Meanwhile, we observed that MBP significantly reduced ATP production in TM3 cells, potentially due to mitochondrial dysfunction. It has been shown that MT could effectively alleviate cadmium-induced mitochondrial dysfunction and reduce nephrotoxicity by inhibiting mitochondrial fission [57]. In line with this finding, our study found that MT inhibited the mitochondrial dynamic imbalance induced by MBP, promoted mitochondrial fusion, and suppressed mitochondrial fission.

PGC-1α regulates mitochondrial dynamics by modulating the expression of key genes. For example, in the muscles of PGC-1α knockout mice, the expression levels of *Mfn1*, *Mfn2*, and *Drp1* were significantly reduced, which resulted in smaller, fragmented mitochondria and a notable decrease in mitochondrial quantity [58]. Moreover, PGC-1α enhances the transcriptional activity of the *Mfn2* promoter, increasing *Mfn2* mRNA levels [59]. Similarly, PGC-1α interacts with the promoter of *Drp1*, thereby regulating its expression [42]. Additionally, upregulation of PGC-1α expression increases the levels of *Mfn1*, *Mfn2*, and *Opa1* while reducing the expression of *Drp1* and *Fis1*, thus maintaining a balance between mitochondrial fusion and fission [60,61,62]. However, the direct relationship between MT-induced upregulation of PGC-1α and mitochondrial dynamics remains unclear. Here, we used SR-18292, a commonly used PGC-1α inhibitor, to suppress the expression of PGC-1α and found that the mRNA level of mitochondrial fission genes significantly increased while those of mitochondrial fusion genes significantly decreased. This suggests that the improvement of mitochondrial dynamics by MT may depend on the regulation of PGC-1α. In future studies, we will further investigate the protein expression of these mitochondrial dynamics-related genes to verify these findings.

It has been demonstrated that MT can concurrently increase the expression of SIRT1 and PGC-1α [42,63]. In this study, we also observed the effect of MT on SIRT1 and PGC-1α expressions in testicular interstitial tissue and TM3 cells. Panes et al. identified a biphasic behavior of the PGC-1α/SIRT1 pathway in the time-dependent toxicity of beta-amyloid peptide in PC12 cells [64]. Under stress conditions, the subcellular distribution of PGC-1α shifts from predominantly nuclear localization to cytoplasmic localization. Meanwhile, cellular stress induces JNK1-mediated phosphorylation of SIRT1, resulting in transient activation of SIRT1 function, significantly diminishing its nuclear/cytoplasmic ratio and displaying behavior similar to cytoplasmic PGC-1α. Furthermore, cellular stress alters the interaction between SIRT1 and PGC-1α [64]. Our study revealed the association between the elevations of SIRT1 and PGC-1α. Inhibition of SIRT1 expression diminished MT’s promoting effect on PGC-1α, while inhibition of PGC-1α curtailed MT’s enhancement of SIRT1 expression, indicating a possible interaction between SIRT1 and PGC-1α. Although this interaction is intriguing, it remains preliminary. Further experiments, such as co-immunoprecipitation, are anticipated to better confirm direct interaction. Additionally, this further supports the hypothesis that MT exerts its effects not only through a single pathway but also by coordinating interactions among multiple molecules to confer cellular protection. However, the specific mechanisms warrant further investigation.

This study has some limitations. First, although TM3 cells provide a valuable in vitro model for investigating Leydig cell biology, they may not fully replicate the steroidogenic capabilities or signaling pathways present in primary Leydig cells observed within the testicular environment. This may limit the direct applicability of our findings to in vivo scenarios related to male reproductive health. Future research should focus on validating these findings through studies involving primary Leydig cells or animal models to better understand the implications for male reproductive health. Second, the methodology employed in our animal experiments did not involve direct exposure of mice to MBP; rather, we administered BPA, from which MBP is metabolized. However, this approach limits our ability to evaluate the direct effects of MBP on the systemic and tissue levels in the animal model. In future studies, we will administer MBP directly to mice to elucidate the effects of MBP at both systemic and tissue levels, thereby enhancing our understanding of the protective role of MT against MBP-induced testicular damage.

## 4. Materials and Methods

### 4.1. Study Animals

Sexually mature male Kunming mice (*N* = 30), each weighing 35 g ± 2 g, were purchased from Yisi Experimental Animals Technology Co., Ltd. (Changchun, China). The mice were housed in a temperature-controlled environment (22 ± 2 °C) with a light/dark cycle of 12 h of light and 12 h of darkness. They had unrestricted access to water and food. The treatment protocol for the mice received approval from the Ethics Committee of Jilin Medical College.

### 4.2. Preparation of Drugs

BPA (#239658, Sigma-Aldrich, St. Louis, MO, USA) was dissolved in olive oil to achieve a concentration of 20 mg/mL. The MT solution for the animal experiments was prepared by dissolving 0.24 g of MT (#M5250, Sigma-Aldrich) in 3 mL of anhydrous ethanol, followed by the addition of 27 mL of normal saline to create an 8 mg/mL MT solution. This solution was further diluted with normal saline to achieve a final concentration of 4 mg/mL. The ethanol content in the MT solution was less than 10% and is considered non-toxic [65,66,67]. For cell experiments, MBP and MT solutions were prepared by dissolving MBP (#S706213, Sigma-Aldrich) or MT (#M5250, Sigma-Aldrich) in 1 mL of DMSO to create a 100 mM stock solution, which was subsequently diluted to various working concentrations using DMEM/F12 medium (#11320033, Gibco, Grand Island, NY, USA). Additionally, SR-18292 (#HY-101491, MedChemExpress, Elizabeth, NJ, USA) and EX-527 (#HY-15452, MedChemExpress) were dissolved in DMSO to create a 50 mM stock solution, which was then diluted to a working concentration of 10 μM using DMEM/F12 medium.

### 4.3. Animal Treatment and Sampling

Mice were randomly divided into the control group, BPA group, and BPA + MT group (N = 10 each). Mice in the BPA group received 50 mg/kg BPA daily via gavage, a dosage considered the lowest observed adverse effect level [68]. Mice in the BPA + MT group were administered 50 mg/kg BPA daily via gavage and were intraperitoneally injected with 20 mg/kg MT every other day [4], while mice in the BPA group received an equivalent volume of saline every other day. Mice in the control group received an equivalent volume of olive oil daily via gavage and were injected with saline every other day. The treatment was conducted for 30 days. After 30 days, the testes were collected from the mice and preserved in Bouin’s solution (BL-GO16, SenBeiJia Biological Technology Co., Ltd., Nanjing, China).

### 4.4. Cell Culture

The TM3 cell line was obtained from Shanghai Guyan Industrial Co., Ltd. (Shanghai, China) and cultured in DMEM/F12 medium (#11320033, Gibco) supplemented with 10% fetal bovine serum (#BL201A, Biosharp, Beijing, China) and 1% penicillin–streptomycin (#BL505A, Biosharp) in an incubator maintained at 37 °C with 5% CO_2_. Cells were passaged upon reaching 80% confluence, with subculturing conducted every 48 h.

### 4.5. CCK-8 Assay

TM3 cells were seeded in a 96-well plate at a density of 5000 cells per well, with seven replicates. After incubation for 12 h, the medium was replaced with DMEM/F12 containing 0.5 μM, 1 μM, 5 μM, 10 μM, 20 μM, and 40 μM MBP, or DMSO, 1 μM, 5 μM, 10 μM, and 25 μM MT. Following drug intervention for 24 h, 20 μL of CCK-8 solution (#C0038, Beyotime, Shanghai, China) was added to each well and incubated for 1 h. Finally, the absorbance of each well was measured at 450 nm using a microplate reader (CMax Plus, Molecular Devices, Shanghai, China). Cell viability was calculated with the following formula: Cell viability (%) = (OD value of experimental group − OD value of blank control group)/(OD value of control group − OD value of blank control group) × 100%. The IC50 (half-maximal inhibitory concentration) value of MBP was determined as the logarithm of drug concentration corresponding to a cell viability of 50%.

### 4.6. Cell Grouping and Treatment

TM3 cells were divided into the control group, DMSO group, MBP group, MT + MBP group, MT + MBP + SR-18292 group, and MT + MBP + EX-527 group. During the MT pre-treatment period (6 h) and the drug intervention period (24 h), cells in the control group were cultured in the DMEM/F12 medium, while those in the DMSO group were cultured in the DMEM/F12 medium for 6 h and then in the presence of 0.04% DMSO for 24 h. Cells in the MBP group were cultured in DMEM/F12 medium for 6 h and then treated with 13 μM MBP for 24 h. The MT + MBP group underwent pre-treatment with 5 μM MT for 6 h, followed by 24 h of treatment with 13 μM MBP. Cells in the MBP + MT + SR-18292 group were initially pre-treated with 5 μM MT for 6 h and subsequently intervened for 24 h with 13 μM MBP and 10 μM SR-18292 [69,70]. In the MBP + MT + EX-527 group, after treatment with 5 μM MT for 6 h, cells were treated for 24 h with 13 μM MBP and 10 μM EX-527 [26,71]. The pre-treatment period of 6 h for MT was determined through pilot experiments. Additionally, cells were treated with 5 μM MT for 1, 3, 6, and 8 h, and Western blotting was used to detect the expression of Sirt1 to determine the optimal treatment duration for MT in promoting Sirt1 expression.

### 4.7. Lactate Dehydrogenase (LDH) Activity Assay

TM3 cells were seeded in a 96-well cell culture plate at a density of 5000 cells per well and divided into the control, DMSO, MBP, and MT + MBP groups. The cell treatment is described above. Seven replicates were set up in each group. The supernatants from each well were transferred to a new 96-well plate, to which the LDH detection working solution (#C0016, Beyotime) was added and incubated in the dark at room temperature for 30 min. Absorbance was measured at 490 nm using a microplate reader. The LDH activity in the samples was calculated (mU/mL), with elevated LDH activity indicating increased drug toxicity.

### 4.8. ELISA

TM3 cells were collected and disrupted using an ultrasonic homogenizer (JY92-IIDN, Ningbo Xinzhi Biotechnology Co., Ltd., Ningbo, China). The samples were centrifuged at 3000 rpm for 20 min at 4 °C, and the supernatants were collected. The levels of testosterone, CYP11A1, and StAR in the supernatant were quantified using ELISA kits for testosterone (#YJ001948), CYP11A1 (#YJ212183), and StAR (#YJ963054) from Shanghai Yuanju Biotechnology Center (Shanghai, China), following the provided protocols. Absorbance values were measured at a wavelength of 450 nm using a microplate reader (CMax Plus, Molecular Devices). The testosterone, CYP11A1, and StAR levels were calculated based on the standard curves.

### 4.9. Reactive Oxygen Species (ROS) Detection

TM3 cells were washed with sterile PBS and incubated with 0.5 mL of DCFH-DA (10 μM/L) fluorescent probe for 30 min. Subsequently, the cells were washed three times with serum-free DMEM/F12 medium and observed under an inverted fluorescence microscope (IX73, OLYMPUS, Tokyo, Japan). Cells exhibiting green fluorescence were identified as ROS-positive. ImageJ v1.8.0 software quantified the ratio of ROS-positive cells, which was calculated as the number of ROS-positive cells divided by the total cell count.

### 4.10. MMP Detection

TM3 cells from each group were incubated with 10 μg/mL of JC-1 probe (#A3516, APExBIO, Houston, TX, USA) at 37 °C in a dark environment for 20 min. Following two washes with sterile PBS, the samples were observed using an inverted fluorescence microscope (IX73, OLYMPUS). At low MMP, the JC-1 probe exists in a monomeric form, emitting green fluorescence, which indicates that the cells are apoptotic; conversely, at high MMP, the JC-1 probe forms polymers that emit red fluorescence, suggesting that the cells are healthy. The ratio of red to green fluorescence was calculated.

### 4.11. TUNEL Assay

DNA fragmentation was assessed using the TUNEL apoptosis detection kit (#MK1019, Boster Biological Technology, Wuhan, China). In detail, TM3 cells were fixed at room temperature with 4% paraformaldehyde for 30 min. For paraffin-embedded tissue samples, the samples were digested with Proteinase K at 37 °C for 15 min. After that, the cell and tissue samples were treated with terminal deoxynucleotidyl transferase and digoxigenin-labeled dUTP for 2 h, followed by a 30 min incubation with 5% BSA at room temperature. Then, after a 30 min reaction at 37 °C with the biotinylated digoxigenin antibody, the cell and tissue samples were incubated with FITC-labeled streptavidin at 37 °C for 30 min. Finally, Hoechst 333258 (#C1018, Beyotime) was added for a 5 min incubation at room temperature to stain the cell nuclei. Cells were then examined under an IX73 microscope (OLYMPUS, Tokyo, Japan), with TUNEL-positive cells emitting bright yellow/green fluorescence. The fluorescence intensity of the cells was analyzed using ImageJ software.

### 4.12. Western Blotting Analysis

TM3 cells were collected and lysed on ice using 150 μL of RIPA lysis buffer (#P0013B, Beyotime) containing protease inhibitors (#K1007, APExBIO). The lysate was centrifuged at 15,000 rpm for 20 min at 4 °C, and the supernatant was collected to determine the protein concentration using a BCA protein assay kit (#P0012S, Beyotime). Equal aliquots of protein were separated by 8–12% SDS-PAGE (#P0012A, Beyotime) and subsequently transferred to a PVDF membrane (#IPVH00010, Millipore, Bedford, MA, USA). The membrane was then blocked with 5% non-fat milk for 1 h and incubated overnight at 4 °C with primary antibodies: PGC-1α (1:1000; ABclonal, Wuhan, China; #A20995), NRF2 (1:1000, ABclonal, #A25327), NRF1 (1:1000, ABclonal, #A3252), SIRT1 (1:1000, ABclonal, #A0230), Bcl-2 (1:300, BIOSS, Beijing, China; #bs-4563R), cleaved-caspase3 (1:1000, #A27145, ABclonal), Bax (1:1000, BIOSS, #bs-0127R), and β-actin (1:100,000, ABclonal, #AC026). After washing, the membrane was incubated with a goat anti-rabbit horseradish peroxidase-conjugated secondary antibody (#A0208, Beyotime) at room temperature for 1 h. Chemiluminescent detection was performed using BeyoECL Plus (#P0018S, Beyotime), and images were captured using the ChemiDOC XRS+ imaging system (Bio-Rad Laboratories, Hercules, CA, USA). The relative expression levels of PGC-1A, NRF2, NRF1, SIRT1, cleaved-caspase3, Bcl-2, Bax, and β-actin were analyzed with ImageJ software.

### 4.13. Quantitative Real-Time PCR (qRT-PCR)

Cellular RNAs were isolated using Trizol (#15596026CN, Invitrogen, Carlsbad, CA, USA), followed by cDNA synthesis using TransScript^®^ One-Step gDNA Removal and cDNA Synthesis SuperMix (#AT311, TransGen Biotech, Beijing, China). qRT-PCR was performed using the ChamQ Universal SYBR qPCR Master Mix (#Q711, Vazyme Biotech, Nanjing, China) on the LightCycler96 PCR instrument (Roche, Basel, Switzerland). The primers were synthesized by Sangon Biological Engineering Technology & Services Co., Ltd. (Shanghai, China), and their sequences are presented in Table 1. A 20 μL reaction mixture was prepared, including 10 μL of 2 × ChamQ Universal SYBR qPCR Master Mix, 0.8 μL of primers (10 μM), and 9.2 μL of template cDNA. The procedures involved pre-denaturation at 95 °C for 30 s, denaturation at 95 °C for 10 s, and annealing and extension at 60 °C for 30 s. The reaction included a total of 40 cycles. The relative expression levels of target genes were calculated using the 2^−ΔΔCt^ method, with GAPDH serving as the housekeeping gene.

### 4.14. EdU Assay

TM3 cells were incubated with 0.5 mL of the EdU working solution (20 μM, #C0075S, Beyotime) for 2 h. Then, the cells were fixed at room temperature using 4% paraformaldehyde for 15 min, and permeabilization was performed with 0.5% Triton X-100 for 10 min. Following this, 0.5 mL of the Click reaction solution was added for incubation in the dark at room temperature for 30 min. The nuclei were stained with Hoechst 333258 (#C1018, Beyotime). The cells were observed and photographed using an inverted fluorescence microscope (IX73, Olympus). The red fluorescence indicated proliferating cells, and the ratio of EdU-positive cells was calculated as the number of EdU-positive cells divided by the total cell count.

### 4.15. Malondialdehyde (MDA) Content Measurement

TM3 cells were lyzed with 150 µL of RIPA lysis buffer on ice for 30 min. After centrifugation at 15,000 rpm for 20 min, the supernatant was collected. Using the MDA detection kit (#S0131S, Beyotime), thiobarbituric acid (TBA) was added to the supernatant and heated in a boiling water bath for 15 min, allowing the MDA to react with TBA, forming a red MDA-TBA adduct. The absorbance of the sample was measured at 532 nm using a microplate reader (CMax Plus, Molecular Devices), and the concentration of MDA in the sample was calculated. The protein concentration of the sample was then determined using the Bradford protein concentration assay kit (#P0006, Beyotime), and the MDA concentration was converted to mol/mg of protein.

### 4.16. ATP Content Detection

The ATP content in cells was measured using an ATP detection kit (#S0026, Beyotime). Briefly, cells were collected and then incubated with 1 mL of ATP lysis buffer (containing firefly luciferase and luciferin) for 2 min at 4 °C. In the presence of ATP, luciferase catalyzed the conversion of luciferin to produce fluorescence. The fluorescence intensity was measured using the FLUOstar OPTIMA microplate reader (BMG LABTECH, Ortenberg, Germany), and the ATP concentration was calculated based on a standard curve. The protein concentration of the sample was determined using a Bradford protein concentration assay kit (#P0006, Beyotime), and the ATP concentration was expressed in μmol/mg of protein.

### 4.17. Immunofluorescence Analysis

Paraffinized testicular tissues were sectioned into 5 μm slices. After deparaffinization, the slices were treated in EDTA antigen retrieval solution (#RA0023, BIOSS) in a boiling water bath for 10 min, followed by a 30 min blocking incubation with 5% BSA. Rabbit anti-SIRT1 primary antibody (1:50, #A0230, ABclonal) was added and incubated overnight at 4 °C. After washing, the slices were incubated with ABflo^®^ 405-conjugated goat anti-rabbit IgG (H + L) (1:200, #AS056, ABclonal) for 1 h in the dark, followed by DAPI nuclear staining for 5 min. Images were captured using a BX53 fluorescence microscope (Olympus), and the average fluorescence intensity of SIRT1 was analyzed with ImageJ software.

### 4.18. Immunohistochemistry

The testicular tissue sections were incubated with 3% H_2_O_2_ at room temperature for 10 min to quench endogenous peroxidase activity. Subsequently, the sections were treated in EDTA antigen retrieval solution (#RA0023, BIOSS) in a boiling water bath for 10 min. Blocking was performed with 5% BSA for 30 min, followed by incubation with rabbit anti-PGC-1α primary antibody (#A20995, ABclonal, diluted 1:50) overnight at 4 °C. After washing, biotin-labeled goat anti-rabbit IgG was applied for 30 min at 37 °C. After washing again, Streptavidin–Biotin Complex (#SA1022, Boster Biological Technology) was added and incubated at 37 °C for 30 min, followed by the DAB (#AR1027, Boster Biological Technology) chromogenic reaction and hematoxylin (#C0107, Beyotime) counterstaining. The samples were observed and photographed using a BX53 microscope (Olympus), and the average optical density of PGC-1α was analyzed with ImageJ software.

### 4.19. Statistical Analysis

Statistical analyses were performed using SPSS 26.0 software. Each experiment was conducted with three independent replicates. The data are presented as mean ± standard deviation. Differences among multiple groups were compared using one-way ANOVA with LSD post hoc tests, with *p* < 0.05 considered statistically significant.

## 5. Conclusions

This study emphasizes that MT plays a significant protective role against oxidative stress and apoptosis in TM3 cells induced by MBP. These protective effects are closely related to the SIRT1/PGC-1α pathway. These findings highlight the role of MT as a potential therapeutic agent for improving male reproductive health compromised by environmental toxins. Further research is warranted to elucidate the detailed mechanisms of MT and to explore its long-term effects in various models of male infertility.

## Figures and Tables

**Figure 1 ijms-26-05910-f001:**
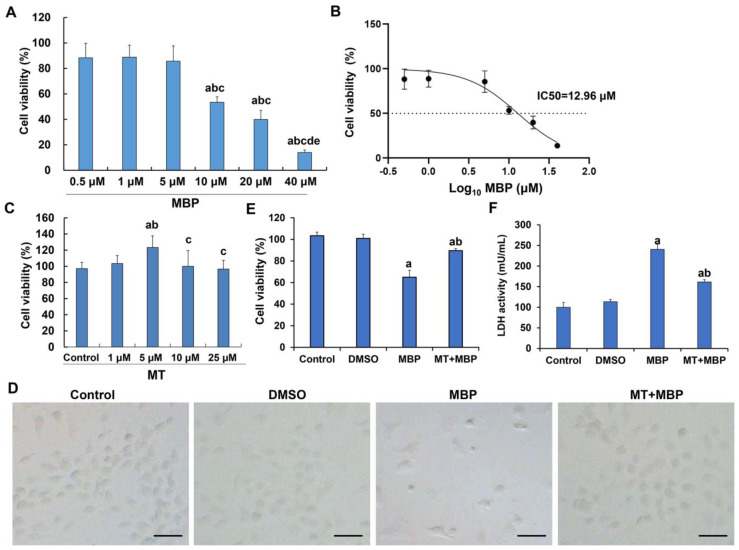
Effect of MBP and MT on TM3 cell viability. (**A**) The cell viability by different concentrations of MBP (*N* = 3). ^a^ Compared to the 0.5 μM MBP group, *p* < 0.05; ^b^ Compared to the 1 μM MBP group, *p* < 0.05; ^c^ Compared to the 5 μM MBP group, *p* < 0.05; ^d^ Compared to the 10 μM MBP group, *p* < 0.05; ^e^ Compared to the 20 μM MBP group, *p* < 0.05. (**B**) The cell viability after MBP treatment (*N* = 3). The X-axis represents the logarithm to the base 10 of the MBP concentrations (μM). The IC50 (half-inhibitory concentration) of MBP was calculated. (**C**) The cell viability after treatment with different concentrations of MT (*N* = 3). ^a^ Compared to the control group, *p* < 0.05; ^b^ Compared to the 1 μM MT group, *p* < 0.05; ^c^ Compared to the 5 μM MT group, *p* < 0.05. (**D**) Cell morphology in different treatment groups. Scale bar: 100 μm. (**E**) The cell viability in different groups (*N* = 3). ^a^ Compared to the control group, *p* < 0.05; ^b^ Compared to the MBP group, *p* < 0.05. (**F**) LDH activity in different groups (*N* = 3). ^a^ Compared to the control group, *p* < 0.05; ^b^ Compared to the MBP group, *p* < 0.05.

**Figure 2 ijms-26-05910-f002:**
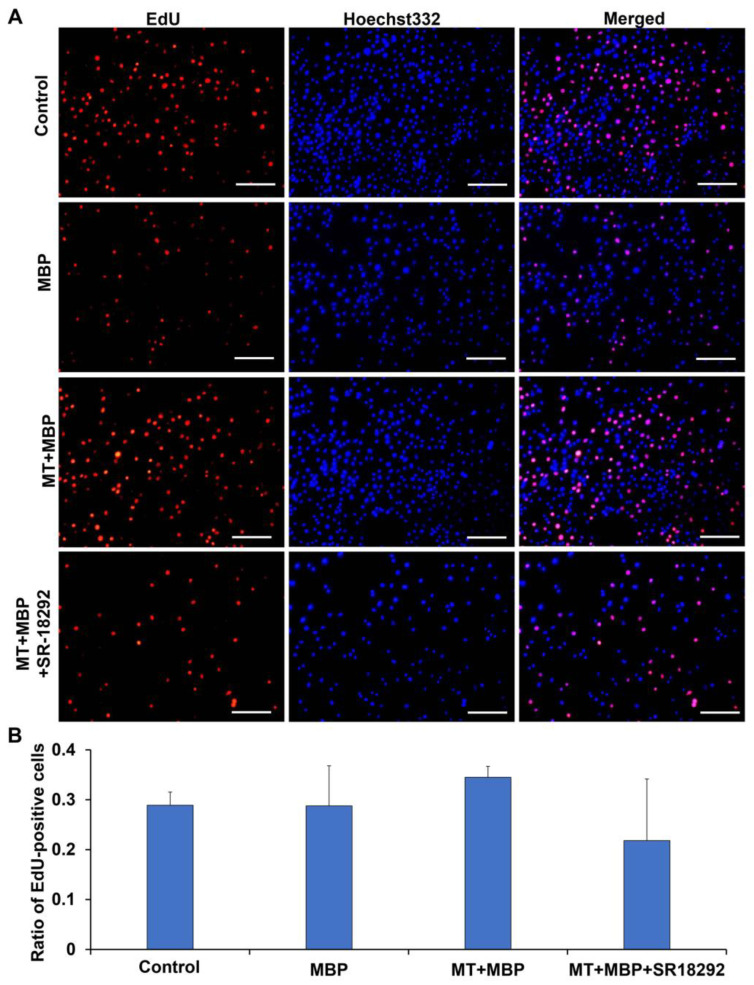
EdU staining in TM3 cells. (**A**) EdU staining of TM3 cells. Red fluorescence denotes EdU-positive TM3 cells, while blue represents the nuclei of TM3 cells. Scale bar: 150 μm. (**B**) The ratio of EdU-positive cells (*N* = 3).

**Figure 3 ijms-26-05910-f003:**
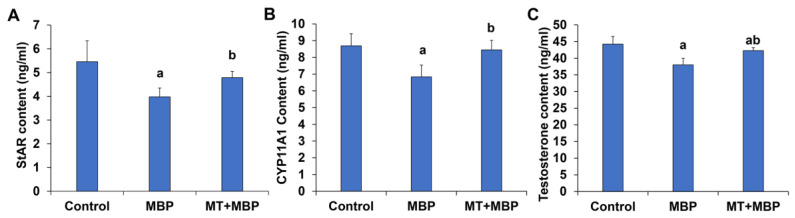
ELISA detection of StAR, CYP11A1, and testosterone levels. (**A**) StAR levels (*N* = 3). (**B**) CYP11A1 levels (*N* = 3). (**C**) Testosterone levels (*N* = 3). ^a^ Compared with the control group, *p* < 0.05; ^b^ Compared with the MBP group, *p* < 0.05.

**Figure 4 ijms-26-05910-f004:**
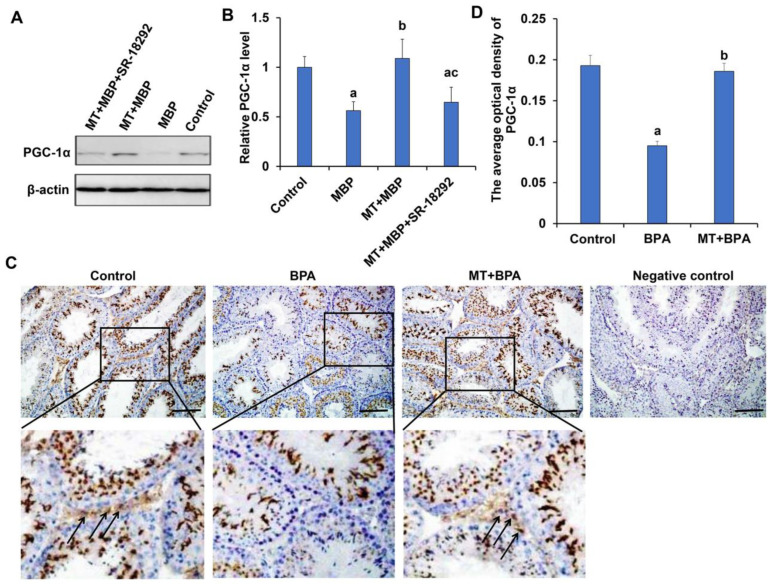
PGC-1α expression in TM3 cells and testicular interstitial tissue. (**A**) Western blotting detection of PGC-1α expression in TM3 cells. (**B**) Relative expression analysis of PGC-1α protein (*N* = 3). ^a^ Compared with the control group, *p* < 0.05; ^b^ Compared with the MBP group, *p* < 0.05. ^c^ Compared to the MT + MBP group, *p* < 0.05. (**C**) Immunohistochemical detection of PGC-1α expression in testicular interstitial tissue. The arrows indicate the expression of PGC-1α in the testicular interstitial tissue. Scale bar: 100 μm. (**D**) Analysis of the average optical density value of PGC-1α protein (*N* = 3). ^a^ Compared with the control group, *p* < 0.05; ^b^ Compared with the BPA group, *p* < 0.05.

**Figure 5 ijms-26-05910-f005:**
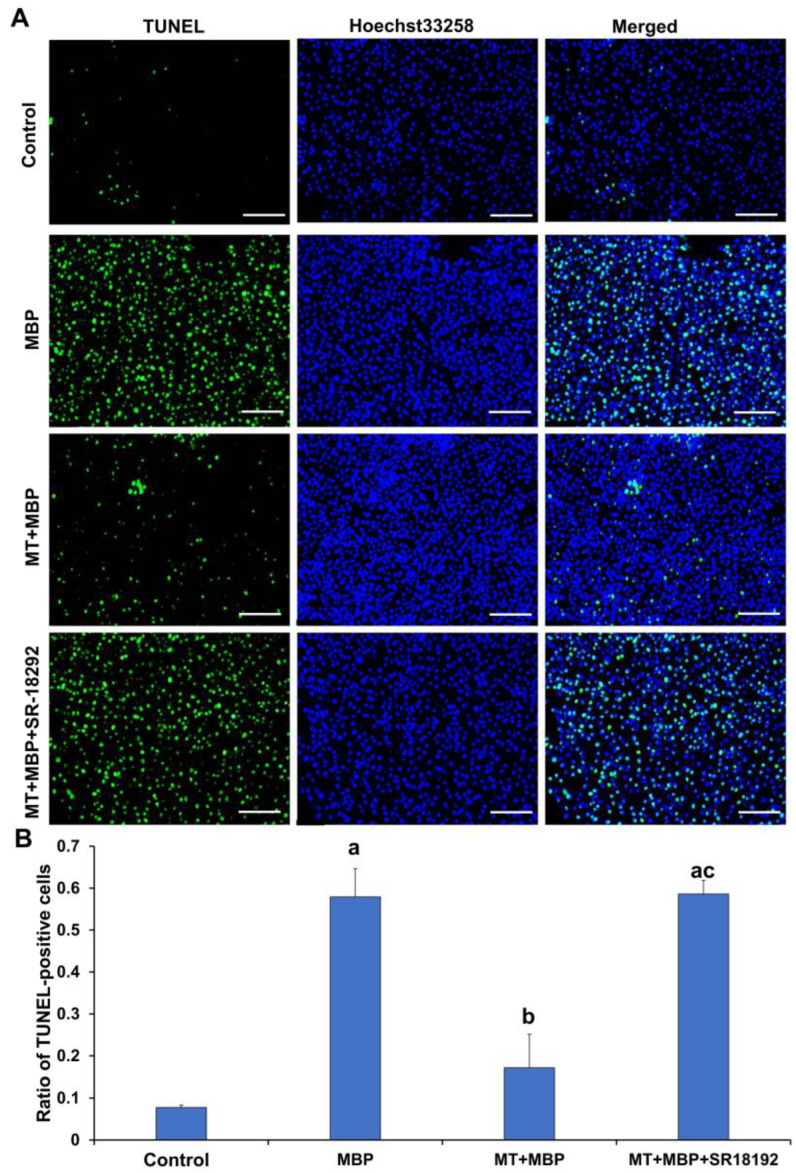
Analysis of apoptosis in TM3 cells. (**A**) TUNEL assay detects apoptosis in TM3 cells. Green fluorescence indicates TUNEL-positive TM3 cells, while blue denotes the nuclei of TM3 cells. Scale bar: 150 μm. (**B**) The ratio of TUNEL-positive cells (*N* = 3). ^a^ Compared with the control group, *p* < 0.05; ^b^ Compared with the MBP group, *p* < 0.05; ^c^ Compared with the MT + MBP group, *p* < 0.05.

**Figure 6 ijms-26-05910-f006:**
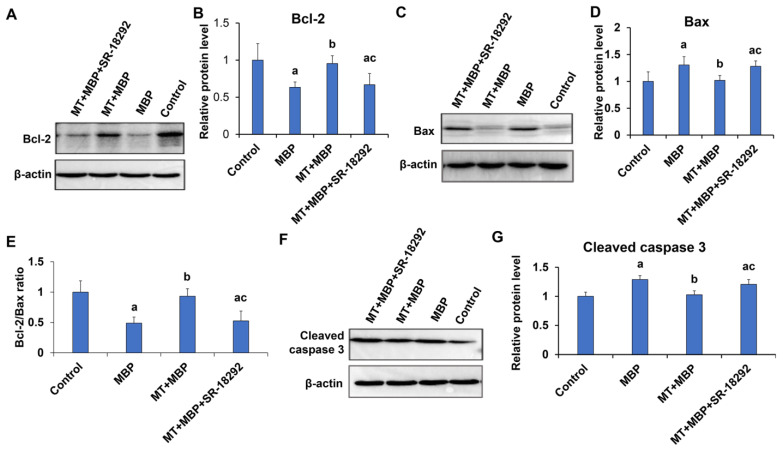
Detection of apoptosis-related protein expression in TM3 cells. (**A**) Western blotting detects Bcl-2 protein expression in TM3 cells. (**B**) Relative expression of Bcl-2 protein in TM3 cells (*N* = 3). (**C**) Western blotting detects Bax protein expression in TM3 cells. (**D**) Relative expression of Bax protein in TM3 cells (*N* = 3). (**E**) The Bcl-2/Bax ratio (*N* = 3). (**F**) Western blotting detects cleaved caspase 3 protein expression in TM3 cells. (**G**) Relative expression of cleaved caspase 3 protein in TM3 cells (*N* = 3). ^a^ Compared with the control group, *p* < 0.05; ^b^ Compared with the MBP group, *p* < 0.05; ^c^ Compared with the MT + MBP group, *p* < 0.05.

**Figure 7 ijms-26-05910-f007:**
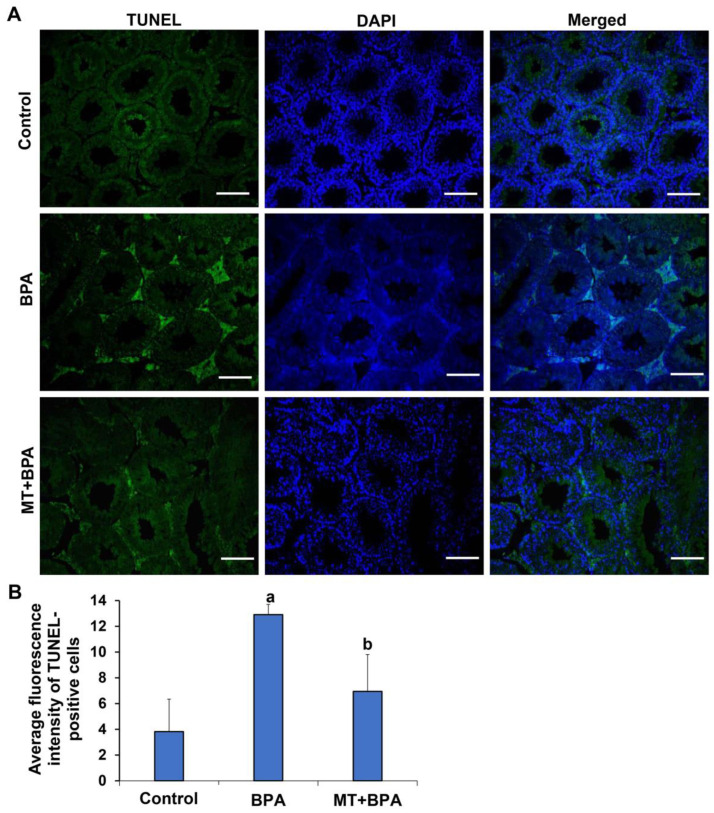
Apoptosis detection in testicular interstitial tissue. (**A**) TUNEL assay detects apoptosis in testicular interstitial tissues. Green fluorescence indicates TUNEL-positive TM3 cells, while blue fluorescence shows the nuclei of TM3 cells. Scale bar: 100 μm. (**B**) Average fluorescence intensity of TUNEL positive cells (*N* = 3). ^a^ Compared with the control group, *p* < 0.05; ^b^ Compared with the BPA group, *p* < 0.05.

**Figure 8 ijms-26-05910-f008:**
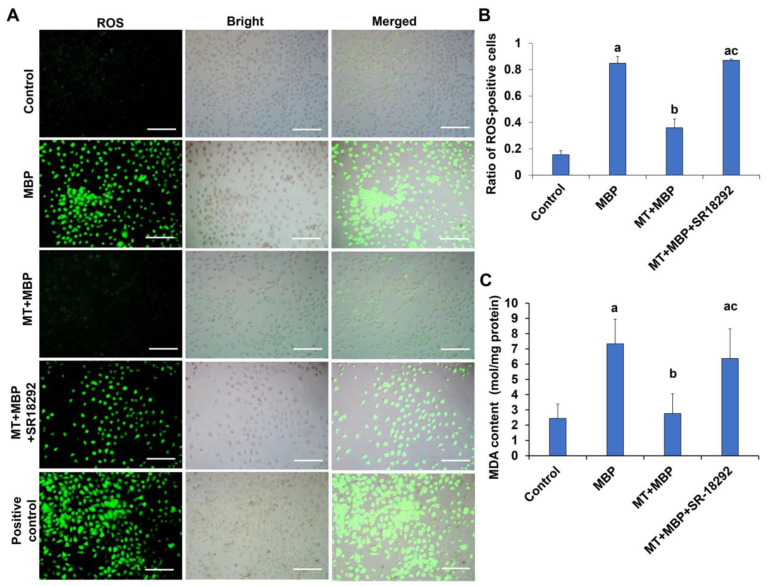
Detection of ROS and MDA in TM3 cells. (**A**) ROS staining in TM3 cells. Green fluorescence indicates ROS-positive TM3 cells. Scale bar: 150 μm. (**B**) The ratio of ROS-positive cells (*N* = 3). (**C**) Detection of MDA levels in TM3 cells (*N* = 3). ^a^ Compared with the control group, *p* < 0.05; ^b^ Compared with the MBP group, *p* < 0.05; ^c^ Compared with the MT + MBP group, *p* < 0.05.

**Figure 9 ijms-26-05910-f009:**
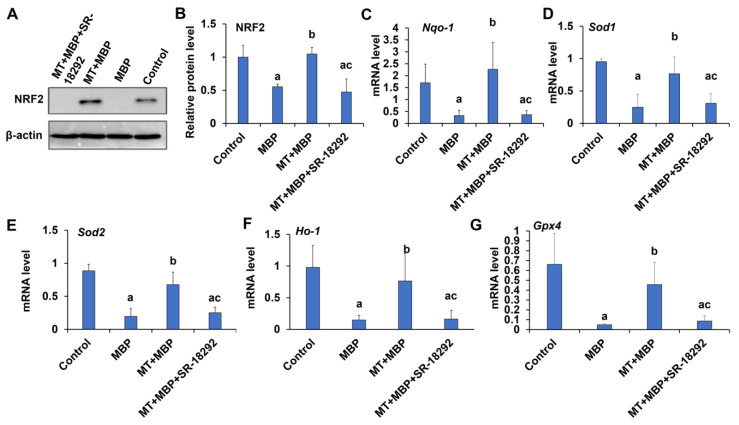
Detection of NRF2 and its downstream antioxidant gene expression. (**A**) Western blotting detects NRF2 protein expression in TM3 cells. (**B**) Relative expression of NRF2 protein in TM3 cells (*N* = 3). qRT-PCR detects *Nqo*-*1* (**C**), *Sod1* (**D**), *Sod2* (**E**), *Ho*-*1* (**F**), and *Gpx4* (**G**) expressions in TM3 cells (*N* = 3). ^a^ Compared with the control group, *p* < 0.05; ^b^ Compared with the MBP group, *p* < 0.05; ^c^ Compared with the MT + MBP group, *p* < 0.05.

**Figure 10 ijms-26-05910-f010:**
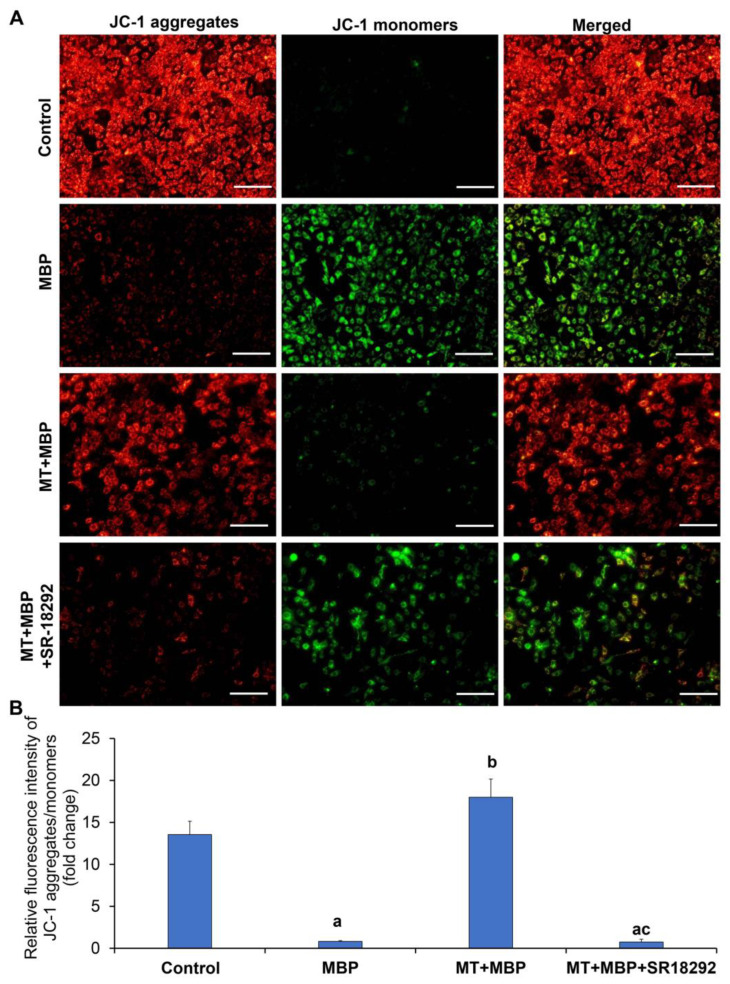
Detection of mitochondrial membrane potential. (**A**) The mitochondrial membrane potential in TM3 cells is assessed after staining with the JC-1 probe. Red fluorescence indicates JC-1 aggregates in TM3 cells, while green fluorescence shows JC-1 monomers in TM3 cells. Scale bar: 150 μm. (**B**) Relative fluorescence intensity of JC-1 aggregates and monomers (*N* = 3). ^a^ Compared to the control group, *p* < 0.05; ^b^ Compared to the MBP group, *p* < 0.05; ^c^ Compared to the MT + MBP group, *p* < 0.05.

**Figure 11 ijms-26-05910-f011:**
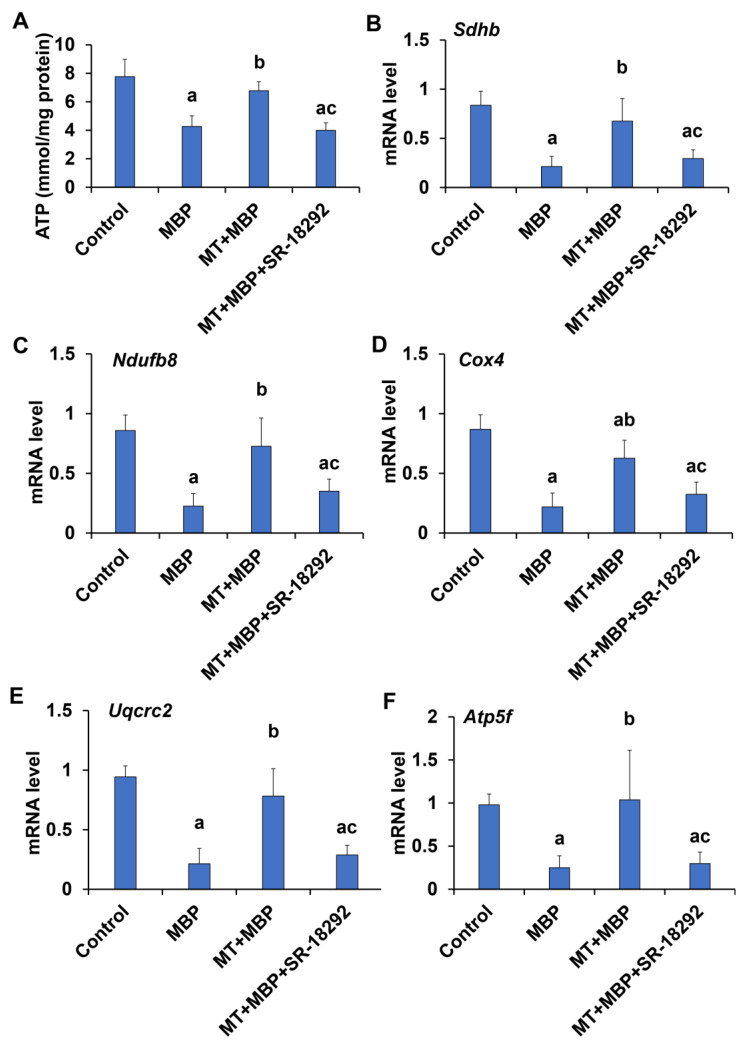
Detection of ATP and mitochondrial respiratory chain complex-related genes. (**A**) ATP production in TM3 cells (*N* = 3). Expression of *Sdhb* (**B**), *Ndufb8* (**C**), *Cox4* (**D**), *Uqcrc2* (**E**), and *Atp5f* (**F**) in TM3 cells, assessed by qRT-PCR (*N* = 3). ^a^ Compared to the control group, *p* < 0.05; ^b^ Compared to the MBP group, *p* < 0.05; ^c^ Compared to the MT + MBP group, *p* < 0.05.

**Figure 12 ijms-26-05910-f012:**
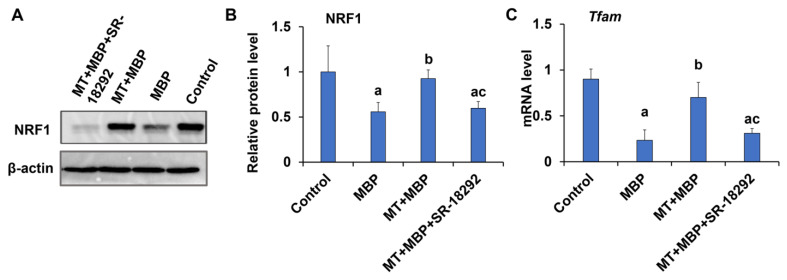
Detection of expression of key mitochondrial biogenesis genes NRF1 and *Tfam*. (**A**) Western blotting analysis of NRF1 protein expression in TM3 cells. (**B**) Relative expression of NRF1 protein in TM3 cells (*N* = 3). (**C**) qRT-PCR analysis of *Tfam* expression in TM3 cells (*N* = 3). ^a^ Compared to the control group, *p* < 0.05; ^b^ Compared to the MBP group, *p* < 0.05; ^c^ Compared to the MT + MBP group, *p* < 0.05.

**Figure 13 ijms-26-05910-f013:**
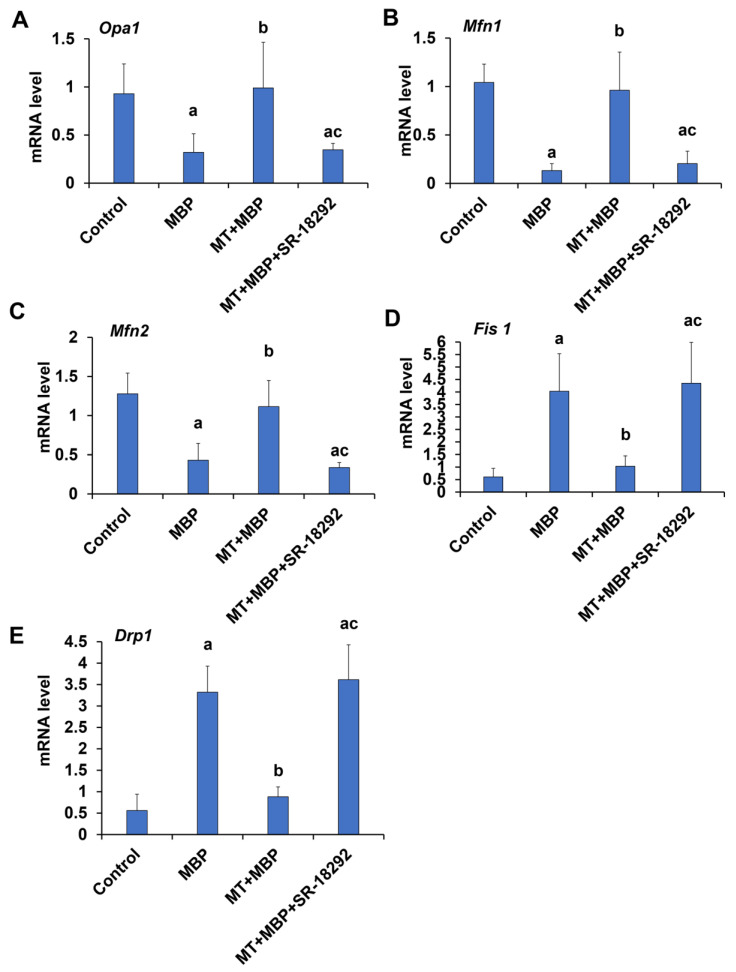
Detection of expression of genes related to mitochondrial dynamics. The gene expression in TM3 cells was measured using qRT-PCR (*N* = 3). The relative mRNA levels of *Opa1* (**A**), *Mfn1* (**B**), *Mfn2* (**C**), *Fis 1* (**D**), and *Drp1* (**E**) are presented. ^a^ Compared to the control group, *p* < 0.05; ^b^ Compared to the MBP group, *p* < 0.05; ^c^ Compared to the MT + MBP group, *p* < 0.05.

**Figure 14 ijms-26-05910-f014:**
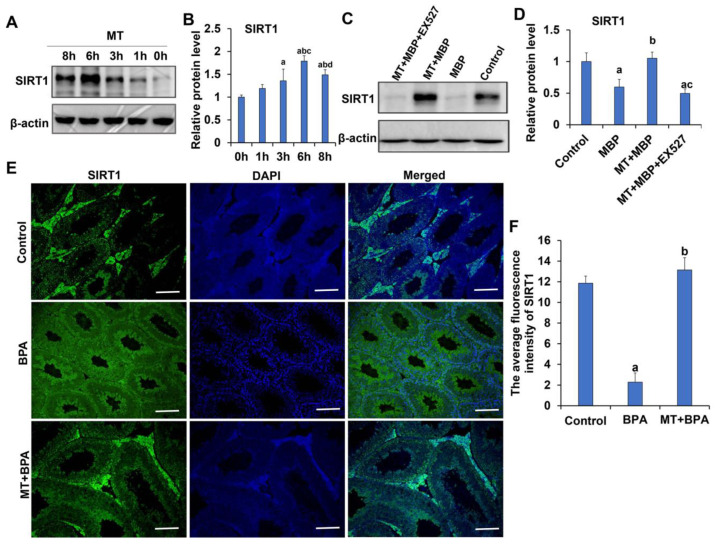
SIRT1 expression in TM3 cells and testicular interstitial tissue. (**A**) Western blotting analysis of SIRT1 expression in TM3 cells treated with different durations of MT. (**B**) Relative expression analysis of SIRT1 protein following varying durations of MT treatment (*N* = 3). ^a^ Compared to the control group, *p* < 0.05; ^b^ Compared to the 1 h MT treatment group, *p* < 0.05; ^c^ Compared to the 3 h MT treatment group, *p* < 0.05; ^d^ Compared to the 6 h MT treatment group, *p* < 0.05. (**C**) Western blotting analysis of SIRT1 expression in TM3 cells across different treatment groups. (**D**) Relative expression analysis of SIRT1 protein in different treatment groups (*N* = 3). ^a^ Compared to the control group, *p* < 0.05; ^b^ Compared to the MBP group, *p* < 0.05. ^c^ Compared to the MT + MBP group, *p* < 0.05. (**E**) Immunofluorescence detection of SIRT1 expression in testicular interstitial tissue. Green fluorescence indicates TUNEL-positive TM3 cells, while blue fluorescence shows the nuclei of TM3 cells. Scale bar: 100 μm. (**F**) Average fluorescence intensity analysis of SIRT1 (*N* = 3). ^a^ Compared to the control group, *p* < 0.05; ^b^ Compared to the BPA group, *p* < 0.05.

**Figure 15 ijms-26-05910-f015:**
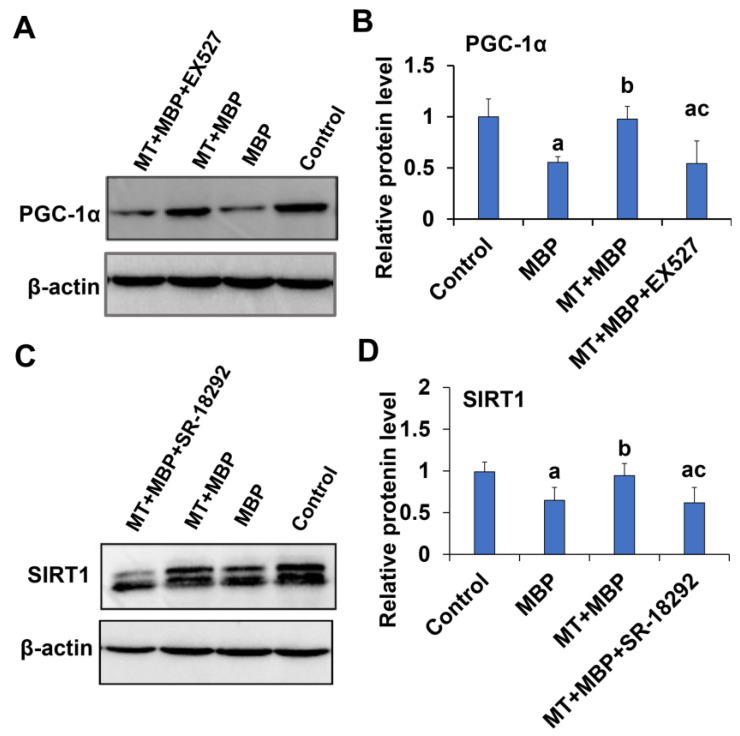
Expression of SIRT1 and PGC-1α in TM3 cells. (**A**) Western blotting analysis of SIRT1 expression in TM3 cells across different treatment groups. (**B**) Relative expression analysis of SIRT1 protein in different treatment groups (*N* = 3). (**C**) Western blotting analysis of PGC-1α expression in TM3 cells across different treatment groups. (**D**) Relative expression analysis of PGC-1α protein in different treatment groups (*N* = 3). ^a^ Compared to the control group, *p* < 0.05; ^b^ Compared to the MBP group, *p* < 0.05; ^c^ Compared to the MT + MBP group, *p* < 0.05.

**Table 1 ijms-26-05910-t001:** Primer sequences.

Genes	Primer Sequence		Annealing Temperature (°C)	Gene ID
*Opa1*	Forward	ATACTGGGATCTGCTGTTGG	59	74143
	Reverse	AAGTCAGGCACAATCCACTT		
*Mfn1*	Forward	GTGTATCTCGCAGTCAGCAGTG	58	67414
	Reverse	TCTTCCCTCTCTTCCATTGAATAAACC		
*Mfn2*	Forward	GCTCAGAAGAGAAGAAGAGTGTCAAG	59	170731
	Reverse	TCCATCAGCACGAGGTCATCC		
*Drp1*	Forward	AACAGGCAACTGGAGAGGAA	56	74006
	Reverse	GCAACTGGAACTGGCACAT		
*Fis1*	Forward	AGCTGGAACGCCTGATTGAT	60	66437
	Reverse	TGGAGACAGCCAGTCCAATG		
*Sod2*	Forward	CAGACCTGCCTTACGACTATGG	58	20656
	Reverse	CTCGGTGGCGTTGAGATTGTT		
*Sod1*	Forward	GGAAGCATGGCGATGAAAGC	59	20655
	Reverse	GCCTTCTGCTCGAAGTGGAT		
*Gpx4*	Forward	CCCGATATGCTGAGTGTGGTTTAC	60	625249
	Reverse	ATTTCTTGATTACTTCCTGGCTCCTG		
*Ho*-*1*	Forward	AGGTACACATCCAAGCCGAGA	58	15368
	Reverse	CATCACCAGCTTAAAGCCTTCT		
*Nqo*-*1*	Forward	AGGATGGGAGGTACTCGAATC	57	18104
	Reverse	TGCTAGAGATGACTCGGAAGG		
*Uqcrc2*	Forward	GACTCTGGGCTCTTTGGA	56	67003
	Reverse	TGTTCTTGGCAGCTTGG		
*Cox4*	Forward	CCCATCCCTCATACTTTCG	59	12857
	Reverse	TCATTCTTGTCATAGTCCCAC		
*Atp5f*	Forward	TCTCCATGCCTCTAACACTCG	59	11946
	Reverse	CCAGGTCAACAGACGTGTCAG		
*Sdhb*	Forward	GAGGGCAAGCAACAGTAT	60	67680
	Reverse	GTCTCCGTTCCACCAGTA		
*Ndufb8*	Forward	GGGACCACTCAGAACTCAGGATG	59	67264
	Reverse	AAAGCCACAAAGCCGAAGAGATG		
*Tfam*	Forward	ATGTGGAGCGTGCTAAAAG	55	21780
	Reverse	ATAGACGAGGGGATGCGAC		
*Gapdh*	Forward	GCACCGTCAAGGCTGAGAAC	59	14433
	Reverse	ATGGTGGTGAAGACGCCAGT		

Note: *Opa1*, optic atrophy 1; *Mfn1*, Mitofusin 1; *Mfn2*, Mitofusin 2; *Drp1*, dynamin-related protein 1; *Fis1*, mitochondrial fission 1; *Sod2*, superoxide dismutase 2; *Sod1*, superoxide dismutase 1; *Gpx4*, glutathione peroxidase 4; *Ho*-*1*, heme oxygenase 1; *Nqo*-*1*, NAD(P)H quinone dehydrogenase 1; *Uqcrc2*, ubiquinol-cytochrome c reductase core protein II; *Cox4*, cytochrome c oxidase subunit 4; *Atp5f*, ATP synthase beta subunit precursor; *Sdhb*, succinate dehydrogenase complex iron-sulfur subunit B; *Ndufb8*, NADH dehydrogenase (ubiquinone) 1 beta subcomplex subunit 8; *Tfam*, mitochondrial transcription factor A; *Gapdh,* glyceraldehyde-3-phosphate dehydrogenase.

## Data Availability

The raw data supporting the conclusions of this article will be made available by the authors on request.

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
