# Peer review of "Melatonin Alleviates MBP-Induced Oxidative Stress and Apoptosis in TM3 Cells via the SIRT1/PGC-1α Signaling Pathway"

_ijms, 2025, doi:10.3390/ijms26125910_

Round 1

Reviewer 1 Report

Comments and Suggestions for Authors

The authors aim to explore the role of melatonin in alleviating the oxidative stress and apoptosis of Leydig cells induced by MBP and the underlying mechanism. Previously, the authors have investigated the MT effect on oxidative stress damage caused by BPA in testicular cells as well as Leydig cells. MBP is the primary active metabolite of BPA, presumably they may have similar functions. Therefore the novelty of this study raise major concern.

  1. Compared to BPA induced oxidative stress and apoptosis, what is the difference of MT effect on those oxidative stress and apoptosis induced by MBP?
  2. In the mice experiment, lack of results indicating the MT effect on the damage induced by MBP at the body and tissue levels. For example, how about the serum or plasma testosterone levels after MT intervened in MBP induced mice, and the WB results of SIRT1/PGC-1α pathway genes in testicular interstitial tissue as well as testicular tissue. The difference of testicular interstitial tissue and testicular tissue need to be clarified, since their functions are different.
  3. Line 70: The reference is incorrect. Please check the reference throughout the manuscript.

Author Response

Comment 1. Compared to BPA induced oxidative stress and apoptosis, what is the difference of MT effect on those oxidative stress and apoptosis induced by MBP?

Response: Thank you for your insightful question regarding the comparison between the effects of MT on oxidative stress and apoptosis induced by BPA versus MBP. We appreciate your concern regarding the novelty of our study.

In our previous research [1], we demonstrated that BPA exposure led to significant oxidative stress and apoptosis in testicular cells, which provided a foundation for understanding the harmful effects of this endocrine disruptor. However, it is essential to recognize that MBP, as the primary active metabolite of BPA, may exhibit unique properties that influence its interactions with cellular pathways in Leydig cells.

In this study, we specifically aimed to explore the effects of MBP on Leydig cells, a key cell type in testicular function and testosterone production, to elucidate the specific mechanisms by which MBP induces oxidative stress and apoptosis. While both BPA and MBP can induce oxidative stress, the underlying pathways may differ due to the structural variations between the two compounds. For instance, MBP may result in distinct alterations in mitochondrial dynamics or differential activation of cell signaling pathways compared to BPA.

Furthermore, MBP is not typically encountered in isolation in vivo, as it is metabolized from BPA within the body. Therefore, examining the direct effects of MBP on Leydig cells allows us to gain insights into the specific molecular mechanisms and cellular responses triggered by this metabolite, which could deviate from those induced by BPA itself.

To clarify these points, we have elaborated on the differences in the mechanisms of action between BPA and MBP in the introduction section of the revised manuscript. We aim to emphasize the significance of studying MBP to advance our understanding of its role as a toxic metabolite and its potential implications for male reproductive health.

Reference:

[1] Qi Q, Yang JX, Li S, Liu JJ, Xu D, Wang GQ, et al. Melatonin alleviates oxidative stress damage in mouse testes induced by bisphenol A. Front Cell Dev Biol. 2024;12:1338828.

Comment 2. In the mice experiment, lack of results indicating the MT effect on the damage induced by MBP at the body and tissue levels. For example, how about the serum or plasma testosterone levels after MT intervened in MBP induced mice, and the WB results of SIRT1/PGC-1α pathway genes in testicular interstitial tissue as well as testicular tissue. The difference of testicular interstitial tissue and testicular tissue need to be clarified, since their functions are different.

Response: In our initial experimental design, we did not directly expose mice to MBP; instead, we administered MBP to TM3 cells. This decision was based on the understanding that MBP, as a metabolite of BPA, is not typically encountered in everyday life. Conversely, BPA is a widely recognized environmental contaminant that can be metabolized into MBP within the body. BPA exposure reflects the real-world scenario where BPA is metabolized into MBP within the body, thus affecting various biological systems. Studying BPA in vivo allows us to examine its systemic effects and the complex interactions that occur in a living organism, as well as to analyze the potential therapeutic effects of MT in mitigating these effects Consequently, the direct effect of MBP on mice and the impact of MT on MBP-induced damage at both the systemic and tissue levels have not been directly investigated. Instead, we used MBP in the in vitro experiments because it is the active metabolite of BPA that exerts direct effects on Leydig cells. By utilizing MBP, we aimed to evaluate the specific cellular and molecular mechanisms of oxidative stress and apoptosis that occur as a direct consequence of metabolic activation of BPA.

However, in light of the importance of scientific rigor and completeness, we acknowledge the necessity of investigating MT’s protective effects against MBP-induced testicular damage in future studies. As suggested by the reviewer, we will directly expose mice to MBP and then assess serum testosterone levels, as well as the expression of SIRT1 and PGC-1α pathway genes in both testicular interstitial tissue and testicular tissue.

To address this comment, we have discussed this in the study limitations section of the revised manuscript.

Additionally, we have verified and clarified the differences between testicular interstitial tissue and testicular tissue following the reviewer’s comments, as reflected in the revised manuscript.

Comment 3. Line 70: The reference is incorrect. Please check the reference throughout the manuscript.

Response: Thank you for bringing this to our attention. In the revised manuscript, we have corrected the reference on line 70 and made any necessary adjustments to other references as well. Additionally, we conducted a comprehensive check of all references throughout the manuscript to ensure that they are correctly cited and align with the formatting guidelines of the journal.

Reviewer 2 Report

Comments and Suggestions for Authors

This study explores the protective effect of melatonin on Leydig cells subjected to MBP-induced oxidative stress and apoptosis, emphasizing the involvement of the SIRT1/PGC-1α signaling axis. The manuscript integrates a variety of molecular, cellular, and biochemical assays to support the proposed mechanisms. This is an interesting paper with clinical implications for male reproductive health and environmental toxicology. However, several specific revisions are required before it becomes qualified to be published. The comments are as follows:

Introduction

Comment #1   The authors used MBP for in vitro and BPA for in vivo experiments. The rationale for this distinction should be clarified, and it would be helpful to explain the physiological relevance of this approach.

Comment #2   Line 65 – 68 on Page 2, the specific cell types utilized in the studies should be explicitly stated in the sentence “MT has been shown to improve oxidative stress, mitochondrial dysfunction…induced by BPA [17]”.

Results

Comment #3   TM3 cells are a practical model for Leydig cell research, but they may differ significantly from primary Leydig cells in steroidogenesis and signaling. Interchangeably using “TM3 cells” and “Leydig cells” should be avoided, please revise throughout the manuscript. It is suggested to explicitly address these limitations when linking findings to reproductive health outcomes.

Comment #4   Multiple figure panels use a dense set of symbols (*, #, â–³, â–¡) to denote significance between various groups. This can be visually overwhelming. The authors are encouraged to adopt letter-based notation (a, b, c) for clarity if needed.

Comment #5   Regarding Section 2.1.1 and Figure 1, the use of “cell growth” to describe single time-point CCK-8 data is imprecise, as proliferation assessment requires multiple time-point detection. The authors should clarify the distinctions between the terms “cell growth”, “cell viability”, and “cell proliferation” to ensure accurate interpretation of results. For instance, while Figure 1 demonstrates the effect of MT on cell viability, this was misinterpreted as the effect on cell growth (see Line 102 – 105 on Page 3).

Comment #6   It is recommended to calculate the Bcl-2/Bax ratio and add the results to Figure 5, as this balance critically determines cellular survival or death.

Comment #7   The placement of Section 2.1.4 ("Effects of MT and MBP on Leydig cell proliferation") disrupts the logical flow between apoptosis and oxidative stress mechanisms. Consider moving the proliferation data subsequent to the cell viability data.

Comment #8   The link between PGC-1α activation and mitochondrial fusion/fission needs further elaboration. Protein expression for OPA1, Mfn1/2, or Drp1 (rather than mRNA alone) would better support the conclusions.

Comment #9   The interaction between SIRT1 and PGC-1α (Figure 15) is intriguing but preliminary. Further experiments such as Co-IP are anticipated to better confirm direct interaction.

Discussion

Comment #10 Limitations of current studies and future plans were not elucidated in Discussion.

Comment #11 Line 505 on Page 27, is it appropriate to use “strong association” here?

Materials and Methods

Comment #12 Section 4.6. (Line 567 – 579 on Page 28 – 29): The rationale for the melatonin pretreatment time should be clarified, specifying whether it was derived from preliminary data, pharmacokinetic analyses, or previous literature. During the pre-treatment period of 6 h, how the control group and the DMSO group were treated should be stated to enhance the comparability of different treatment groups.

Comment #13 The authors set up “seven duplicate wells” when conducting LDH activity assay, please explain why setting so many? In addition, “duplicate” often refers to repeating twice, so “seven replicates” would be more appropriate in this context. Please also add the information of how many replicates were conducted for each assay.

Comment #14 Please pay attention to the rules of gene symbol formatting for mouse genes (i.e. start with an uppercase letter, followed by lowercase letters; gene symbols are italicized) and revise accordingly throughout the manuscript.

Other comments

Comment #15 Other minor issues:

The legend of Figure 1 should mention “TM3 cells”.

All “P” should be italicized.

Fix inconsistent formatting (e.g., "PGC-1α" vs. "PGC-1A") (See Line 646 on Page 30, and check throughout the manuscript).

Line 708 on Page 32: “H2O2”.

Comment #16 The manuscript would benefit from language polishing by a native English speaker or professional editing service to improve clarity and flow.

Author Response

Introduction

Comment #1   The authors used MBP for in vitro and BPA for in vivo experiments. The rationale for this distinction should be clarified, and it would be helpful to explain the physiological relevance of this approach.

Response: We appreciate the reviewer’s valuable feedback.

In our study, we chose to use MBP in the in vitro experiments because it is the active metabolite of BPA that exerts direct effects on Leydig cells. By utilizing MBP, we aimed to evaluate the specific cellular and molecular mechanisms of oxidative stress and apoptosis that occur as a direct consequence of metabolic activation of BPA.

Conversely, we utilized BPA in the in vivo experiments as it represents the form to which organisms are typically exposed in the environment. BPA exposure reflects the real-world scenario where BPA is metabolized into MBP within the body, thus affecting various biological systems. Studying BPA in vivo allows us to examine its systemic effects and the complex interactions that occur in a living organism, as well as to analyze the potential therapeutic effects of MT in mitigating these effects.

By using MBP in vitro and BPA in vivo, we elucidated both the intricate cellular mechanisms influenced by the metabolite and the systemic consequences of the parent compound as encountered in real-world conditions. We hope this clarification addresses the reviewer’s concerns effectively, and we have incorporated this explanation into the revised manuscript for added clarity.

Comment #2   Line 65 – 68 on Page 2, the specific cell types utilized in the studies should be explicitly stated in the sentence “MT has been shown to improve oxidative stress, mitochondrial dysfunction…induced by BPA [17]”.

Response: We have revised the mentioned sentence to explicitly state the specific cell types utilized. The revised content is as follows:

“MT has been shown to improve oxidative stress, mitochondrial dysfunction, and apoptosis induced by polybrominated diphenyl ethers through the SIRT1/PGC-1α signaling pathway in fish kidney cells [16], and to alleviate mitochondrial dynamics imbalance and apoptosis in BPA-induced colon injury [17].”

Results

Comment #3   TM3 cells are a practical model for Leydig cell research, but they may differ significantly from primary Leydig cells in steroidogenesis and signaling. Interchangeably using “TM3 cells” and “Leydig cells” should be avoided, please revise throughout the manuscript. It is suggested to explicitly address these limitations when linking findings to reproductive health outcomes.

Response: Thank you for your valuable suggestion. In the revised manuscript, we have clearly distinguished between TM3 cells and primary Leydig cells throughout the text. Specifically, we have avoided using the terms “TM3 cells” and “Leydig cells” interchangeably.

Furthermore, although TM3 cells provide a valuable in vitro model for investigating Leydig cell biology, they may not fully replicate the steroidogenic capabilities or signaling pathways present in primary Leydig cells observed within the testicular environment. This may limit the direct applicability of our findings to in vivo scenarios related to male reproductive health. Future research should focus on validating these findings through studies involving primary Leydig cells or animal models to better understand the implications for male reproductive health. To enhance clarity, we have discussed this in the revised manuscript.

Comment #4   Multiple figure panels use a dense set of symbols (*, #, â–³, â–¡) to denote significance between various groups. This can be visually overwhelming. The authors are encouraged to adopt letter-based notation (a, b, c) for clarity if needed.

Response: As suggested, we have adopted letter-based notation (a, b, c, d) instead of the dense set of symbols (*, #, â–³, â–¡) to denote significance between various groups. We have also updated the figure legends accordingly. Please refer to the revised figures and the revised manuscript for details.

 Comment #5   Regarding Section 2.1.1 and Figure 1, the use of “cell growth” to describe single time-point CCK-8 data is imprecise, as proliferation assessment requires multiple time-point detection. The authors should clarify the distinctions between the terms “cell growth”, “cell viability”, and “cell proliferation” to ensure accurate interpretation of results. For instance, while Figure 1 demonstrates the effect of MT on cell viability, this was misinterpreted as the effect on cell growth (see Line 102 – 105 on Page 3).

Response: We apologize for any confusion caused. In this study, we used a CCK-8 assay to assess cell viability. To enhance clarity, we have corrected the phrasing to accurately reflect that our findings pertain to “cell viability” rather than “cell growth.” Please refer to the revised Figure 1 and the revised manuscript for details.

Comment #6   It is recommended to calculate the Bcl-2/Bax ratio and add the results to Figure 5, as this balance critically determines cellular survival or death.

Response: In response to your recommendation, we have calculated the Bcl-2/Bax ratio for our experimental groups. Our results found that this ratio was significantly reduced in the MBP and MT+MBP+SR-18292 groups, but was significantly elevated in the MT+MBP group. This finding has been included in the results section and the revised figure. We have also updated the figure legends accordingly. Please refer to the revised figure and the revised manuscript for details.

Comment #7   The placement of Section 2.1.4 ("Effects of MT and MBP on Leydig cell proliferation") disrupts the logical flow between apoptosis and oxidative stress mechanisms. Consider moving the proliferation data subsequent to the cell viability data.

Response: As suggested, we have moved Section 2.1.4 (“Effects of MT and MBP on Leydig cell proliferation”) to follow the cell viability data (Section 2.1.1). The figures have been renumbered accordingly. Please kindly review.

 Comment #8   The link between PGC-1α activation and mitochondrial fusion/fission needs further elaboration. Protein expression for OPA1, Mfn1/2, or Drp1 (rather than mRNA alone) would better support the conclusions.

Response: In the revised manuscript, we have added a substantial discussion regarding the link between PGC-1α and mitochondrial dynamics, incorporating relevant studies that support the role of these proteins.

In the initial experimental design, we focused on measuring mRNA levels as we aimed to examine the transcriptional regulation of these mitochondrial dynamics-related proteins by PGC-1α. Unfortunately, we faced limitations due to insufficient sample availability, which currently prevents us from presenting protein expression results in this study. However, we recognize the importance of this data to strengthen our conclusions. Therefore, we have included a statement in the revised manuscript indicating our intention to conduct future experiments to analyze the protein expression levels of OPA1, Mfn1/2, or Drp1.

Comment #9   The interaction between SIRT1 and PGC-1α (Figure 15) is intriguing but preliminary. Further experiments such as Co-IP are anticipated to better confirm direct interaction.

Response: Thank you for your valuable comment regarding the interaction between SIRT1 and PGC-1α. We agree that additional experiments, such as co-immunoprecipitation (Co-IP), would greatly enhance the robustness of our findings by providing direct evidence of this interaction. However, due to limitations in sample availability, we were unable to include Co-IP experiments in this revision. We appreciate your understanding of this limitation, and we have highlighted this in our revised manuscript. We plan to conduct these experiments in the future to further elucidate the interactions between SIRT1 and PGC-1α.

Discussion

Comment #10 Limitations of current studies and future plans were not elucidated in Discussion.

Response: Thanks for the comment. In the revised manuscript, we have added a dedicated section in the Discussion that explicitly highlights the limitations of our study. Additionally, we have outlined our plans for future research to address these limitations and further explore the topics raised in our study.

Please refer to the following information and the revised manuscript for details.

“Limitations and future research directions

This study has some limitations. First, although TM3 cells provide a valuable in vitro model for investigating Leydig cell biology, they may not fully replicate the steroidogenic capabilities or signaling pathways present in primary Leydig cells observed within the testicular environment. This may limit the direct applicability of our findings to in vivo scenarios related to male reproductive health. Future research should focus on validating these findings through studies involving primary Leydig cells or animal models to better understand the implications for male reproductive health. Second, the methodology employed in our animal experiments did not involve direct exposure of mice to MBP; rather, we administered BPA, from which MBP is metabolized. However, this approach limits our ability to evaluate the direct effects of MBP on the systemic and tissue levels in the animal model. In future studies, we will administer MBP directly to mice to elucidate the effects of MBP at both systemic and tissue levels, thereby enhancing our understanding of the protective role of MT against MBP-induced testicular damage.”

Comment #11 Line 505 on Page 27, is it appropriate to use “strong association” here?

Response: We apologize for any confusion caused. In this study, we have indeed made preliminary observations of the interaction between SIRT1 and PGC-1α but did not conduct an in-depth exploration of this relationship. As such, we have revised the wording “strong association” into “the association” to ensure it accurately reflects the current findings.

Materials and Methods

Comment #12 Section 4.6. (Line 567 – 579 on Page 28 – 29): The rationale for the melatonin pretreatment time should be clarified, specifying whether it was derived from preliminary data, pharmacokinetic analyses, or previous literature. During the pre-treatment period of 6 h, how the control group and the DMSO group were treated should be stated to enhance the comparability of different treatment groups.

Response: The decision to use a 6-hour pretreatment period for MT was based on preliminary experiments, which indicated that this duration resulted in the highest expression levels of SIRT1. Additionally, during the 6-hour pretreatment period, the control group and DMSO group were administered DMEM.

To enhance clarity, we have included these details in the revised Materials and Methods section under “Cell Grouping and Treatment”.

Comment #13 The authors set up “seven duplicate wells” when conducting LDH activity assay, please explain why setting so many? In addition, “duplicate” often refers to repeating twice, so “seven replicates” would be more appropriate in this context. Please also add the information of how many replicates were conducted for each assay.

Response: In our LDH activity analysis, we set up seven wells per group to minimize variability arising from differences in culture wells and experimental procedures. It is generally advisable in cell experiments to utilize more than four wells per group to ensure greater reliability of the results. Additionally, each experiment was conducted with three independent replicates.

To improve clarity, we have modified the terminology from “seven duplicate wells” to “seven replicates” in the manuscript. We have also added information regarding the number of replicates conducted for each assay to enhance clarity.

Comment #14 Please pay attention to the rules of gene symbol formatting for mouse genes (i.e. start with an uppercase letter, followed by lowercase letters; gene symbols are italicized) and revise accordingly throughout the manuscript.

Response: As suggested, we have reviewed the entire manuscript to ensure that all gene symbols for mouse genes are formatted correctly. For example, we have changed the initial letter of each gene symbol to uppercase, followed by lowercase letters, and italicized the gene symbols as per the established guidelines.

Other comments

Comment #15 Other minor issues:

The legend of Figure 1 should mention “TM3 cells”.

All “P” should be italicized.

Fix inconsistent formatting (e.g., "PGC-1α" vs. "PGC-1A") (See Line 646 on Page 30, and check throughout the manuscript).

Line 708 on Page 32: “H2O2”.

Response: Thank you for your helpful comments regarding the minor issues in the manuscript. We have made the following revisions in response:

1) Figure Legend: We have revised the legend of Figure 1 to include “TM3 cells.” The updated legend now reads: “Effect of MBP and MT on TM3 cell viability.”

2) Italicization of “P”: We have italicized all instances of “P” throughout the manuscript.

3) Consistent Formatting: We have addressed the inconsistent formatting by changing “PGC-1A” to “PGC-1α” throughout the manuscript.

4) We have revised “H2O2” to “H2O2”.

Comment #16 The manuscript would benefit from language polishing by a native English speaker or professional editing service to improve clarity and flow.

Response: As suggested, we have sought the assistance of a professional editing service to enhance the overall quality of our writing.

Reviewer 3 Report

Comments and Suggestions for Authors

The manuscript entitled “Melatonin alleviates MBP-induced oxidative stress and apoptosis in Leydig cells via the SIRT1/PGC-1a signaling pathway” investigates the potential of melatonin to alleviate the toxic effects of MBP on Leydig cells and the molecular mechanisms implicated. The manuscript is very well written and describes eloquently and sufficiently the experimental work and outcomes. The research is well-designed and the conclusions are well-supported by the data presented.

I have only minor suggestions, which I hope will improve the manuscript.

  1. The full name of MBP and its metabolic origin from BPA should be mentioned in the abstract.
  2. The Introduction is rather short and concise. I would recommend that a short description of the key molecules studied is also included there. In particular, some parts that are included in the Discussion about the role of PGC-1a (see Lines 415-418), SIRT1, and maybe StAR and CYP11A1could be transferred in the Introduction.
  3. Figure 1B, what is shown in the x axis of the graph? What is the Ig in MBP (Ig, μM)?
  4. Figure 1C, can they provide a plausible explanation why out of all concentrations tested, only 5 μM of melatonin had an effect on cell viability?
  5. Figure 3A shows that in the presence of the inhibitor the effect of melatonin on PGC1-1a expression is alleviated. I am not very familiar with this molecule, could the authors please explain how it inhibits the expression of PGC-1a? My understanding is that it inhibits the activity of PGC-1a, not its expression.
  6. Figure 7, the representative images of EdU do not seem to correspond to the data presented in the graph (Fig 7B). The presented images demonstrate a clear decrease in EdU-positive cells in MBP-treated and MT-MBP-SR-1892-treated cells. This is not reflected in the graph. Can the authors provide some insight on this?
  7. Figure 8B, the Y axis corresponds to the ratio of ROS-positive cells. What ratio?
  8. Regarding the in vivo studies, it is not clear how were melatonin and BPA administered time-wise. Was melatonin administered before BPA and how long before? Which were the “alternate days”?

Author Response

Comment 1. The full name of MBP and its metabolic origin from BPA should be mentioned in the abstract.

Response: In response to your suggestion, we have revised the abstract to include the full name of MBP (4-methyl-2,4-bis(4-hydroxyphenyl)pent-1-ene) and its metabolic origin from BPA.

Comment 2. The Introduction is rather short and concise. I would recommend that a short description of the key molecules studied is also included there. In particular, some parts that are included in the Discussion about the role of PGC-1a (see Lines 415-418), SIRT1, and maybe StAR and CYP11A1could be transferred in the Introduction.

Response: As suggested, we have expanded the Introduction and included brief descriptions of the key molecules, such as SIRT1, PGC-1α, StAR, and CYP11A1.

Comment 3. Figure 1B, what is shown in the x axis of the graph? What is the Ig in MBP (Ig, μM)?

Response: To clarify, “lg” in the x-axis of Figure 1B represents the logarithm to the base 10 of the MBP concentrations, and the units for MBP are expressed in micromoles per liter (μM). Therefore, the X-axis of the graph indicates the log-transformed concentrations of MBP (μM). To avoid confusion, we have modified the label of the x-axis into Log10 MBP (μM). Moreover, we have added this information to the figure legend for Figure 1 to enhance clarity.

Comment 4. Figure 1C, can they provide a plausible explanation why out of all concentrations tested, only 5 μM of melatonin had an effect on cell viability?

Response: Thanks for the comment. In our in vitro culture of TM3 cells, the oxygen concentration in the cultivation environment is around 20%, which may induce mild oxidative stress compared to the in vivo environment. At concentrations from 1 μM to 5 μM, melatonin may exert antioxidant properties to mitigate this oxidative stress, resulting in enhanced cell viability. However, further increasing the concentration of melatonin to above 10 μM resulted in a reduction in cell viability. Similarly, Govindasamy et al. found that when Schwann cells were treated with melatonin concentrations ranging from 0.5 to 20 μM for 24 hours, cell viability increased with concentrations up to 10 μM, but decreased at 20 μM [1]. This decline observed in both our study and that of Govindasamy et al. could be attributed to cytotoxic effects associated with higher concentrations of melatonin. We have included this explanation in the second paragraph of the Discussion section to enhance the understanding of our findings.

Reference:

[1] Govindasamy N, Chung Chok K, Ying Ng P, Yian Koh R, Moi Chye S. Melatonin Induced Schwann Cell Proliferation and Dedifferentiation Through NF-ĸB, FAK-Dependent but Src-Independent Pathways. Rep Biochem Mol Biol. 2022 Apr;11(1):63-73. doi: 10.52547/rbmb.11.1.63.

Comment 5. Figure 3A shows that in the presence of the inhibitor the effect of melatonin on PGC1-1a expression is alleviated. I am not very familiar with this molecule, could the authors please explain how it inhibits the expression of PGC-1a? My understanding is that it inhibits the activity of PGC-1a, not its expression.

Response: Thank you for your insightful question. We fully agree with your understanding that the inhibitor SR-18292 primarily influences the activity of PGC-1α rather than directly inhibiting its expression. SR-18292 promotes the acetylation of PGC-1α, thereby suppressing its activity. In the revised Figures 4A-4B (original Figures 3A-3B), our results demonstrated that the expression of PGC-1α in the MT + MBP group was significantly higher than that in the MBP group. However, in the MT + MBP + SR-18292 group, the expression of PGC-1α significantly decreased compared to the MT + MBP group. Interestingly, we also observed a significant reduction in SIRT1 expression in the MT + MBP + SR-18292 group (Figures 15C and 15D). Notably, Yang et al. reported that knockdown of SIRT1 using siRNA significantly decreased PGC-1α expression [1]. Therefore, it is likely that SR-18292 affects PGC-1α expression by regulating SIRT1 levels. The precise mechanism underlying this interaction warrants further investigation.

Reference:

[1] Yang M, Chen X, Zhang M, Zhang X, Xiao D, Xu H, Lu M. hUC-MSC preserves erectile function by restoring mitochondrial mass of penile smooth muscle cells in a rat model of cavernous nerve injury via SIRT1/PGC-1a/TFAM signaling. Biol Res. 2025 Jan 27;58(1):8. doi: 10.1186/s40659-024-00578-y.

Comment 6. Figure 7, the representative images of EdU do not seem to correspond to the data presented in the graph (Fig 7B). The presented images demonstrate a clear decrease in EdU-positive cells in MBP-treated and MT-MBP-SR-1892-treated cells. This is not reflected in the graph. Can the authors provide some insight on this?

Response: Thanks for the comment. In this study, we calculated the ratio of EdU-positive cells, which is the number of EdU-positive cells divided by the total cell count. Due to the effects of MBP on cell viability, the total number of cells in the MBP and MT + MBP + SR-1892 groups decreased, resulting in a reduction in the number of EdU-positive cells as well. However, statistical analysis showed that the ratio of EdU-positive cells did not differ significantly among the groups. To address this issue, we have added further clarification in the EdU assay subsection of the Materials and Methods.

Comment 7. Figure 8B, the Y axis corresponds to the ratio of ROS-positive cells. What ratio?

Response: The ratio of ROS-positive cells refers to the number of ROS-positive cells divided by the total cell count. To address this issue, we have added further clarification in the ROS detection section of the Materials and Methods.

Comment 8. Regarding the in vivo studies, it is not clear how were melatonin and BPA administered time-wise. Was melatonin administered before BPA and how long before? Which were the “alternate days”?

Response: Thanks for the comment. In the in vivo studies, melatonin and BPA were administered simultaneously to the mice. BPA was administered via oral gavage every day, while melatonin was injected into the mice (i.p.) every other day. The design of the in vivo experiments differs from that of the in vitro experiments. This is because if melatonin were administered to the mice before BPA, it might be metabolized, thereby failing to alleviate BPA-induced testicular damage. Therefore, in our in vitro design, melatonin and BPA were used concurrently. In studies [1-3] where melatonin is employed to mitigate testicular damage caused by endocrine disruptors such as Di(2-ethylhexyl) phthalate (DEHP), metformin, and nicotine, it is also administered simultaneously with these testicular-damaging agents. We have added further clarification regarding the use of melatonin and BPA in the “Animal Treatment and Sampling” section of the Materials and Methods. Please refer to the revised manuscript for details.

References:

[1] Bahrami N, Goudarzi M, Hosseinzadeh A, Sabbagh S, Reiter RJ, Mehrzadi S. Evaluating the protective effects of melatonin on di(2-ethylhexyl) phthalate-induced testicular injury in adult mice. Biomed Pharmacother. 2018 Dec;108:515-523. doi: 10.1016/j.biopha.2018.09.044.

[2] Tajabadi E, Javadi A, Azar NA, Najafi M, Shirazi A, Shabeeb D, Musa AE. Radioprotective effect of a combination of melatonin and metformin on mice spermatogenesis: A histological study. Int J Reprod Biomed. 2020 Dec 21;18(12):1073-1080. doi: 10.18502/ijrm.v18i12.8029.

[3] Mohammadghasemi F, Jahromi SK. Melatonin ameliorates testicular damages induced by nicotine in mice. Iran J Basic Med Sci. 2018 Jun;21(6):639-644. doi: 10.22038/IJBMS.2018.28111.6829.

Round 2

Reviewer 1 Report

Comments and Suggestions for Authors

The authors have addressed my comments.